# Blocking Kir6.2 channels with SpTx1 potentiates glucose-stimulated insulin secretion from murine pancreatic β cells and lowers blood glucose in diabetic mice

Yajamana Ramu[†], Jayden Yamakaze[†], Yufeng Zhou, Toshinori Hoshi, Zhe Lu*

Department of Physiology, Perelman School of Medicine University of Pennsylvania, Philadelphia, United States

**Abstract** ATP-sensitive K$^+$ (K$_{ATP}$) channels in pancreatic β cells are comprised of pore-forming subunits (Kir6.2) and modulatory sulfonylurea receptor subunits (SUR1). The ATP sensitivity of these channels enables them to couple metabolic state to insulin secretion in β cells. Antidiabetic sulfonylureas such as glibenclamide target SUR1 and indirectly suppress Kir6.2 activity. Glibenclamide acts as both a primary and a secondary secretagogue to trigger insulin secretion and potentiate glucose-stimulated insulin secretion, respectively. We tested whether blocking Kir6.2 itself causes the same effects as glibenclamide, and found that the Kir6.2 pore-blocking venom toxin SpTx1 acts as a strong secondary, but not a strong primary, secretagogue. SpTx1 triggered a transient rise of plasma insulin and lowered the elevated blood glucose of diabetic mice overexpressing Kir6.2 but did not affect those of nondiabetic mice. This proof-of-concept study suggests that blocking Kir6.2 may serve as an effective treatment for diabetes and other diseases stemming from K$_{ATP}$ hyperactivity that cannot be adequately suppressed with sulfonylureas.

**\*For correspondence:**
zhelu@pennmedicine.upenn.edu

[†]These authors contributed equally to this work.

**Competing interest:** The authors declare that no competing interests exist.

## Editor's evaluation

ATP-sensitive potassium channels (K$_{ATP}$ channels) play a key role in glucose-stimulated insulin secretion from pancreatic β cells, and they are a major therapeutic target in diabetes mellitus. This paper provides clear evidence in mice that the effects of a peptide toxin that blocks the pore-forming Kir6.2 subunit are subtly but importantly different from the effects of sulfonylureas that act on an auxiliary subunit of the K$_{ATP}$ channels. The pore-blocking peptide toxin has effects on β cells and on mice only when they contain a mutant channel that is sensitive to the toxin, unlike the native mouse channel; in contrast, the sulfonylurea glibenclamide appears to have additional downstream effects on insulin secretion, even in the absence of the normal glucose stimulus.

## Introduction

Diabetes mellitus is a group of diseases that all manifest elevated blood glucose levels but with different underlying causes (*American Diabetes Association, 2011*). Among these diseases, neonatal diabetes mellitus (NDM) was traditionally considered a variant in Type 1 diabetes mellitus (T1DM) and had accordingly been treated with insulin. Since the early 2000s, NDM has been recognized as a genetic disorder that stems from gain-of-function mutations in pancreatic ATP-sensitive K$^+$ (K$_{ATP}$) channels (*Gloyn et al., 2004*). The octameric K$_{ATP}$ channel protein complexes, each of which consists of four pore-forming inward-rectifier 6.2 (*Kcnj11* or Kir6.2) subunits and four surrounding auxiliary sulfonylurea receptor 1 (*Abcc8* or SUR1) subunits (*Aguilar-Bryan et al., 1995*; *Inagaki et al., 1995*), are present in

the plasma membrane of β cells within islets of Langerhans in the pancreas. The discovery of the mutations underlying NDM was anticipated by the experimental demonstration in mice that the expression of Kir6.2 with gain-of-function mutations caused hypoinsulinemia and hyperglycemia (*Koster et al., 2000*). Subsequently, this mutation-caused pathological phenomenon was further examined in mice by overexpressing another gain-of-function mutant Kir6.2 known to cause NDM at the time (*Girard et al., 2009*). Mutations in Kir6.2 tend to cause more severe phenotypes of NDM than those in SUR1, and the hyperactivity of the resulting mutant $K_{ATP}$ channels is less likely to be adequately suppressed by sulfonylureas (*Hattersley and Ashcroft, 2005*; *Pipatpolkai et al., 2020*). The severe-phenotype-causing mutations of Kir6.2 are also associated with developmental delay, epilepsy, and permanent neonatal diabetes (DEND syndrome). In this regard, Kir6.2 represents a new potential drug target.

$K_{ATP}$ channels were originally discovered in the plasma membrane of cardiac myocytes (*Noma, 1983*) and later found to exist in many other tissue types. $K_{ATP}$ channels in pancreatic β cells are inhibited by both extracellular glucose and intracellular ATP (*Ashcroft et al., 1984*; *Rorsman and Trube, 1985*). The ATP sensitivity of these channels is thought to contribute to the regulation of insulin secretion from pancreatic β cells in the following manner (*Ashcroft and Rorsman, 2013*; *Ashcroft and Rorsman, 2012*; *Nichols, 2006*). When the blood glucose concentration is low, the overall metabolism in β cells remains low. Consequently, the ratio of intracellular ATP to ADP is relatively low, and the $K_{ATP}$ channels tend to be open, helping to maintain the hyperpolarized resting membrane potential ($V_m$). An elevated blood glucose level increases the metabolism in β cells and thus the ratio of intracellular ATP to ADP. An increase in ATP relative to ADP inhibits $K_{ATP}$ channels, depolarizing $V_m$ and thereby increasing the voltage-gated $Ca^{2+}$ channel ($Ca_V$) activity. An increased $Ca_V$-mediated $Ca^{2+}$ influx raises the concentration of intracellular free $Ca^{2+}$ ($[Ca^{2+}]_{in}$), a signal required for triggering robust exocytotic secretion of insulin.

As a class of the antidiabetic drugs, sulfonylureas act as impactful primary secretagogues, triggering insulin release in the presence of a nonstimulating, basal concentration of glucose. These drugs also act as strong secondary secretagogues, robustly potentiating the insulin secretion stimulated by an elevated concentration of glucose. Sulfonylureas bind to SUR1 and thereby indirectly inhibit currents through the $K_{ATP}$ channels (*Gribble and Reimann, 2003*; *Henquin, 1992*). The commonly used sulfonylurea glibenclamide is membrane permeable and has been shown to lodge inside SUR1, and its effect on insulin secretion cannot be rapidly reversed (*Lee et al., 2017*; *Li et al., 2017*; *Martin et al., 2017*; *Schatz et al., 1977*; *Wu et al., 2018*).

The suppression of $K_{ATP}$ activity undoubtedly contributes to the ability of glibenclamide to promote insulin secretion; however, some studies have suggested that glibenclamide also interacts with other proteins involved in the secretory process, in addition to $K_{ATP}$ channels in the plasma membrane (*Eliasson et al., 1996*; *Hinke, 2009*; *Kang et al., 2011*; *Lehtihet et al., 2003*; *Renström et al., 2002*; *Shibasaki et al., 2014*; *Tian et al., 1998*; *Zhang et al., 2009*). Moreover, while the plasma membrane of β cells is strongly depolarized by raising the concentration of extracellular $K^+$ to 30 mM, an application of glibenclamide under this depolarized condition can still promote additional insulin secretion without further changes in $[Ca^{2+}]_{in}$ (*Geng et al., 2007*). These findings about the glibenclamide action raise the question that to what extent, a direct blockade of the ion-conduction pore of the $K_{ATP}$ channel alone mimics the primary and secondary secretagogue effects of glibenclamide. To address this question, an effective inhibitor of Kir6.2 itself is required.

Our group previously searched for inhibitors of human Kir6.2 (hKir6.2) and discovered that five small proteins in the venoms of certain centipedes inhibited hKir6.2. In particular, a 54-residue protein toxin isolated from the venom of *Scolopendra polymorpha*, dubbed SpTx1, is the most potent inhibitor with a $K_d$ of 15 nM (*Ramu et al., 2018*; *Ramu and Lu, 2019*). Here, using SpTx1, we set out to test the impact of a direct blockade of Kir6.2 on insulin secretion in mice.

## Results

### SpTx1 does not affect the blood glucose levels in wild-type mice

We examined whether SpTx1 had an influence on the fasting blood glucose levels of wild-type mice. The blood glucose level of the overnight-fasted wild-type mice was 173 (±3.18) mg/dl in the following morning (*Figure 1A*). Given that the blood glucose level of wild-type mice can be lowered by glibenclamide (*Remedi and Nichols, 2008*), we used it as a positive control. As expected, a single

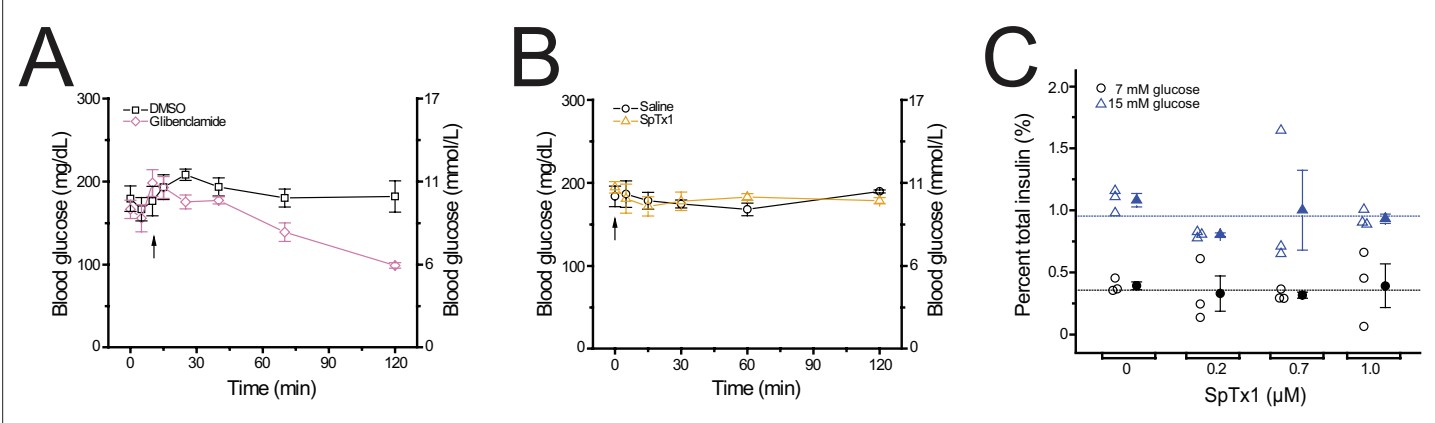

**Figure 1.** SpTx1 does not affect the blood glucose level or potentiate glucose-stimulated insulin secretion (GSIS) from isolated pancreatic islets of wild-type B6J mice. (**A, B**) Blood glucose levels (mean ± standard error of the mean [SEM], $n$ = 5 independent experiments for each dataset) of overnight-fasted mice (8–12 weeks of age) measured at indicated time points during a 2-hr observation period. (**A**) Glibenclamide (40 mg/kg, purple diamonds) or its vehicle DMSO (black squares) was administered by an intraperitoneal injection (arrow). (**B**) SpTx1 (1 mg/kg, orange triangles) or its vehicle saline solution (black circles) was administered by an intravenous injection (arrow). (**C**) Dot plots of GSIS from isolated pancreatic islets of 8- to 12-week-old mice. For each independent experiment under specified conditions, individual groups of five islets were placed in the wells of a microwell plate for insulin release assay. The secreted insulin as a percentage of the total insulin content of the islets from each experiment is plotted against the indicated concentration of SpTx1 in the presence of 7 mM (open black circles) or 15 mM (open blue triangles) glucose, with the mean ± SEM of each group presented to the right of the respective dataset (filled black circles or blue triangles, $n$ = 3 independent sets of experiments). The blue or black horizonal line indicates the average of data for all SpTx1 concentrations in 7 or 15 mM glucose.

The online version of this article includes the following source data for figure 1:

**Source data 1.** Related to *Figure 1A–C*.

intraperitoneal application of glibenclamide (40 mg/kg body weight) lowered the blood glucose level of wild-type mice by half at the end of a 2-hr observation period whereas that of the vehicle dimethyl sulfoxide (DMSO) did not have any meaningful effect. To assess the effects of SpTx1 on the blood glucose levels, intravenous (IV) injection was used to administer SpTx1 to avoid the interpretation of the result being confounded by the toxin's bioavailability. SpTx1 at a dose of 1 mg/kg neither had discernible effect on the blood glucose levels of wild-type mice (*Figure 1B*) nor caused other noticeable differences during the observation period, when compared to wild-type mice administered with the vehicle saline. Using total blood volume, this dose of SpTx1 is calculated to be 3 µM, ~200 times the $K_d$ value of SpTx1 against hKir6.2. The finding that SpTx1, estimated to be at 3 µM in the blood, had no effect on the blood glucose level, suggests that the toxin does not have any consequential actions (in the present context) by either inhibiting mouse Kir6.2 (mKir6.2) channels or acting on unintended targets in wild-type mice.

## SpTx1 fails to potentiate insulin secretion of pancreatic islets from wild-type mice

We tested whether SpTx1 affected the amount of glucose-stimulated insulin secretion (GSIS) from β cells in isolated pancreatic islets of wild-type mice using two different stimulating concentrations of glucose. To perform this test, we adopted a static assay of insulin release from isolated islets incubated in microwell plates (*Remedi et al., 2009*). We used high concentrations of SpTx1 between 0.2 and 1 µM in the bath solutions because there was likely a SpTx1 concentration gradient from the extracellular solution to the interior of an islet. In this concentration range, SpTx1 failed to alter insulin secretion in the presence of either 7 or 15 mM glucose (*Figure 1C*). Thus, SpTx1 had no significant effects on GSIS either through acting on mKir6.2 or any unintended targets within the wild-type mouse islets.

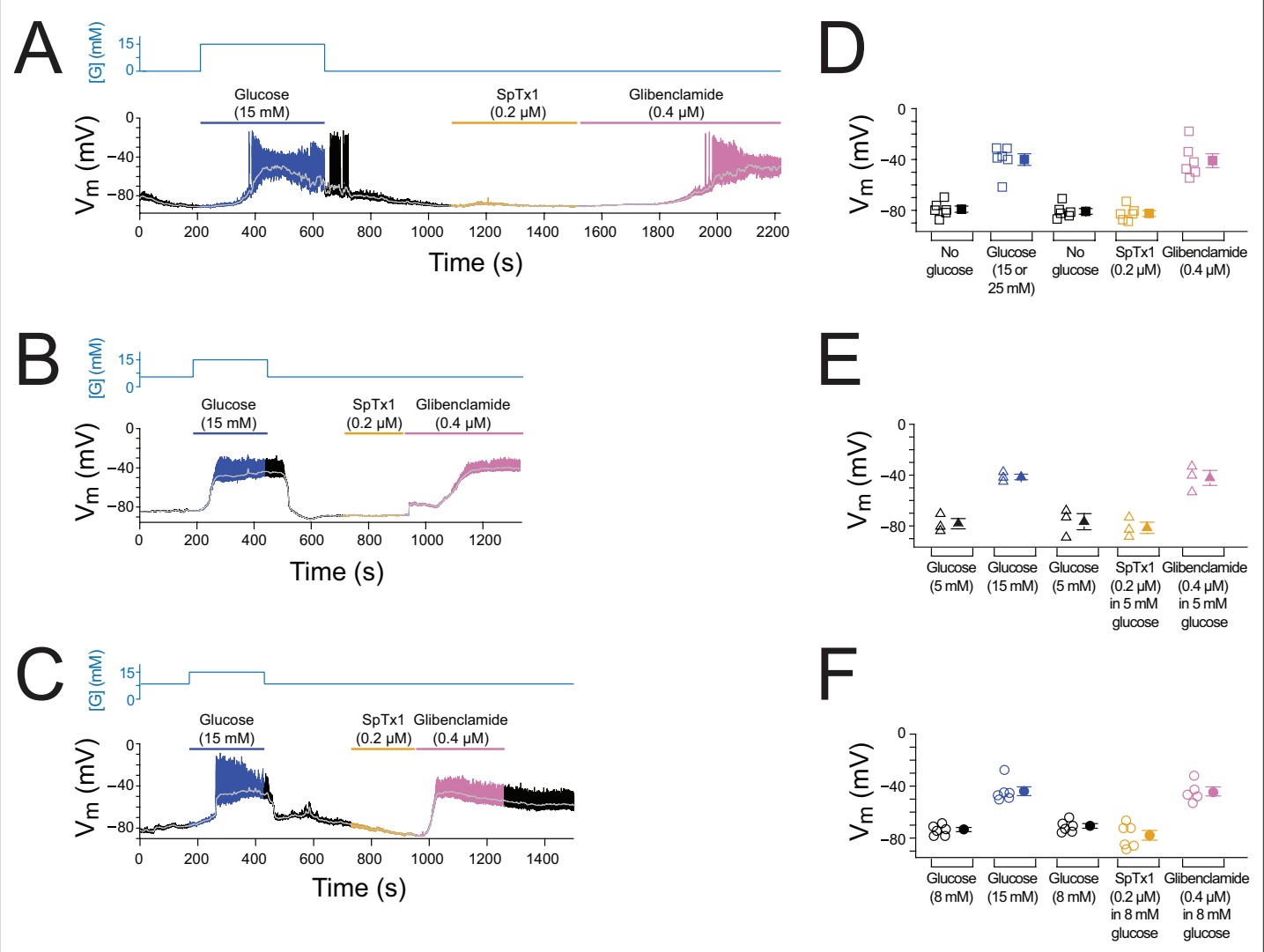

**Figure 2.** SpTx1 does not depolarize the membrane potential ($V_m$) of β cells in isolated pancreatic islets from wild-type mice. (**A–C**) $V_m$ traces recorded in the perforated whole-cell mode from individual β cells near the surface of isolated but intact islets from B6J wild-type mice (8–12 weeks of age). The switching of the glucose concentration from 0 mM (**A**), 5 mM (**B**), or 8 mM (**C**) to 15 mM and back is indicated by the blue schematic line at the top, and the application of 0.2 µM SpTx1 (orange) or 0.4 µM glibenclamide (purple) or the presence of 15 mM glucose in the bath solution (blue) is as indicated by their color-coded lines above the $V_m$ trace. The light gray curve overlaid on the $V_m$ trace was obtained by (offline) filtering of the recorded trace at 0.1 Hz using a low-pass Gaussian routine. (**D–F**) Dot plots of filtered $V_m$ values for individual cells from multiple islets under the conditions of the corresponding illustrative traces shown in (**A–C**), where their mean ± standard error of the mean (SEM) are plotted on the right as filled symbols with errors bars. Squares, $n = 6$ independent experiments, are as plotted in (**D**); triangles, $n = 3$ in (**E**); circles, $n = 5$ or 6 in (**F**). 15 or 25 mM glucose was used and data for these conditons were plotted together in (**D**).

The online version of this article includes the following source data for figure 2:

**Source data 1.** Related to *Figure 2A–F*.

## SpTx1 does not depolarize the membrane potential of β cells in pancreatic islets from wild-type mice

The prevailing excitation–secretion coupling paradigm regarding β cells postulates that a blockade of $K_{ATP}$ channels in pancreatic β cells markedly depolarizes $V_m$ to trigger insulin release. Previous studies reported that dissociated β cells exhibited altered electrophysiological properties, for example, a much greater cell-to-cell variation in glucose sensitivity, compared with the cells in isolated islets (*Salomon and Meda, 1986*; *Scarl et al., 2019*). Thus, $V_m$ responses of individual cells in isolated but intact islets were recorded in the perforated whole-cell mode (*Figure 2*). In a typical mouse islet, up

to 80% of the cells are β cells (*Steiner et al., 2010*). In our measurements, β cells were identified by their characteristic glucose-stimulated $V_m$ depolarization and subsequent bursts of (non-overshooting) action potentials. Illustrative $V_m$ changes in responses to an increase of the extracellular glucose concentration to 15 mM from the starting concentration of 0, 5, or 8 mM are shown in *Figure 2A–C*. For example, increasing the glucose concentration from 0 to 15 mM led to $V_m$ depolarization and a burst of action potentials in a reversible manner (*Figure 2A*, blue and black segments), thus functionally confirming that the recorded cell was a β cell. Subsequent application of SpTx1 at 0.2 μM had little effect on $V_m$ (*Figure 2A*, orange segment) but that of the sulfonylurea glibenclamide at 0.4 μM dramatically depolarized $V_m$ and triggered a train of action potentials (*Figure 2A*, purple segment). Similar results documenting the ineffectiveness of SpTx1 and the contrasting effectiveness of glibenclamide in inducing $V_m$ depolarization were also observed using the starting extracellular glucose concentrations of 5 and 8 mM (*Figure 2B, C*). Low-pass filtered (*Figure 2A–C*, light gray traces) time-averaged $V_m$ values from multiple islets under the three different glucose conditions are summarized in *Figure 2D–F*. Thus, in wild-type mouse islets, glucose and glibenclamide dramatically depolarize $V_m$ but SpTx1, despite its high concentration (>10 times the $K_d$ value for hKir6.2), has no such effect. These results also serve as an additional control study to determine if SpTx1 has impactful off-targets in the islets.

## SpTx1 inhibits hKir6.2 and mKir6.2 with markedly different affinities

The most likely cause for the failure of SpTx1 to depolarize $V_m$ and to potentiate GSIS in wild-type mouse β cells is that SpTx1 does not potently inhibit mKir6.2. Indeed, the amino acid sequences of hKir6.2 and mKir6.2 are extremely similar but not identical (96% identity). To determine whether SpTx1 targets the two $K_{ATP}$ channel orthologs with different affinities, we examined mKir6.2 and hKir6.2 coexpressed with their respective SUR1 in *Xenopus* oocytes so that we could directly compare the effect of SpTx1 on $mK_{ATP}$ and $hK_{ATP}$ channels under the same conditions (*Figure 3*).

As expected, 10 nM SpTx1 suppressed currents through $hK_{ATP}$ channels comprised of hKir6.2 and hSUR1 (hKir6.2 + hSUR1) by about half (*Figure 3A*; *Ramu et al., 2018*). In contrast, a comparable suppression of currents through $mK_{ATP}$ channels, each of which consists of mKir6.2 and mSUR1 (mKir6.2 + mSUR1), required 1 μM SpTx1, 100 times higher concentration (*Figure 3B*). The concentration dependence of current inhibition by SpTx1 was fitted using a model for the toxin-to-channel interaction with one-to-one stoichiometry, yielding an apparent $K_d$ of 15 and 673 nM for $hK_{ATP}$ and $mK_{ATP}$, respectively (*Figure 3E*). For ease of comparison, all apparent $K_d$ values in the entire study are summarized in *Figure 3—figure supplement 1*. Clearly, SpTx1 inhibits $mK_{ATP}$ with a much lower potency than $hK_{ATP}$.

## Differences between Kir6.2 orthologs underlie their different SpTx1 affinities

It is imperative to exclude the possibility that some differences between the auxiliary hSUR1 and mSUR1 subunits are the primary causes of the aforementioned large difference in apparent SpTx1 affinities of the two orthologous channel complexes. First, we compared the effects of SpTx1 on currents through channels that consisted of hKir6.2 + hSUR1, hKir6.2 + mSUR1, mKir6.2 + mSUR1, and mKir6.2 + hSUR1 (*Figure 3A–D*). SpTx1 inhibited currents through hKir6.2 + hSUR1 and hKir6.2 + mSUR1 with high affinities but, in contrast, the toxin inhibited currents through mKir6.2 + mSUR1 and mKir6.2 + hSUR1 with low affinities (*Figure 3E, F*). Second, we examined the Kir6.2 mutant that lacks 26 C-terminal residues, dubbed Kir6.2-ΔC26; unlike wild-type Kir6.2, this mutant forms functional channels on the cell surface without needing coexpression with SUR1 (*Tucker et al., 1997*). For a technical advantage, we used the version of Kir6.2-ΔC26 with an additional mutation at the N-terminal region, V59G, which boosted the apparent expression of the $K_{ATP}$ current (*Proks et al., 2004*). The resulting constitutively active channels with this double (background) mutation are hereafter referred to as Kir6.2[bgd], which carry robust currents in whole oocytes that contain millimolar concentrations of ATP. SpTx1 inhibited currents through hKir6.2[bgd] much more potently than those through mKir6.2[bgd] (*Figure 4A*, blue squares versus orange circles). The above two lines of results indicate that differences in the two pore-forming orthologous Kir6.2 themselves, but not in their SUR1, primarily underlie the observed differential SpTx1 affinities, and are consistent with the idea that SpTx1 inhibits currents through $K_{ATP}$ channels by interacting with Kir6.2 (*Ramu et al., 2018*; *Ramu and Lu, 2019*).

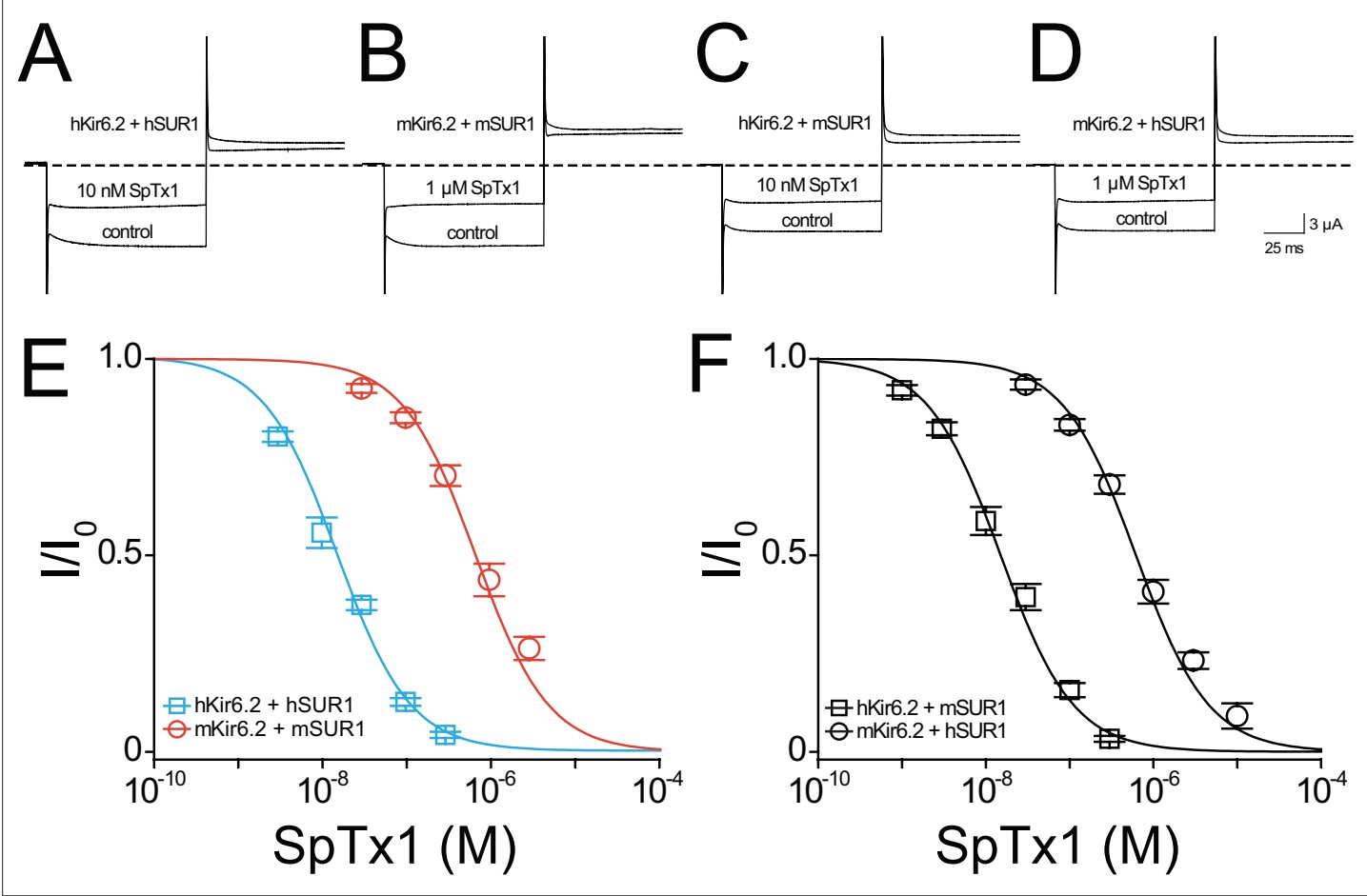

**Figure 3.** SpTx1 inhibits hKir6.2 and mKir6.2 with markedly different affinities. Currents through hKir6.2 coexpressed with hSUR1 (hK$_{ATP}$; **A**), mKir6.2 with mSUR1 (mK$_{ATP}$; **B**), hKir6.2 with mSUR1 (**C**), and mKir6.2 with hSUR1 (**D**) activated by adding 3 mM azide to the 100 mM K$^+$-containing bath solution and recorded in the absence (control) or presence of 10 nM (**A, C**) or 1 µM (**B, D**) SpTx1. The currents were elicited by stepping voltages from the holding potential of 0 to −80 mV and then to +80 mV. The dashed line indicates zero-current level. (**E, F**) Fractions of remaining channel currents ($I/I_o$) plotted against the concentration of SpTx1. The curves superimposed on data correspond to the fits of an equation for a bimolecular reaction. The fitted $K_d$ values are 1.53 (±0.13) × 10$^{-8}$ M for hKir6.2 coexpressed with hSUR1 (**E**, cyan squares), 6.27 (±1.02) × 10$^{-7}$ M for mKir6.2 with mSUR1 (**E**, vermilion circles), 1.47 (±0.14) × 10$^{-8}$ M for hKir6.2 with mSUR1 (**F**, black squares), and 6.44 (±0.69) × 10$^{-7}$ M for mKir6.2 with hSUR1 (**F**, black circles), where data are plotted as mean ± standard error of the mean (SEM; $n$ = 5 independent experiments).

The online version of this article includes the following source data and figure supplement(s) for figure 3:

**Source data 1.** Related to *Figure 3A–F*.

**Figure supplement 1.** SpTx1 affinities of Kir6.2 and of its variants in the study.

**Figure supplement 1—source data 1.** Related to *Figure 3—figure supplement 1*.

Furthermore, the low affinity of SpTx1 for mKir6.2 explains this toxin's observed ineffectiveness in influencing GSIS from isolated islets (*Figure 1C*) and $V_m$ in individual β cells (*Figure 2*).

## Variation at a single residue causes different affinities of hKir6.2 and mKir6.2 for SpTx1

Next, we proceeded to identify the residue variation causing the different SpTx1 sensitivities in hKir6.2 and mKir6.2. Many venom proteins target the external vestibule of K$^+$ channels including Kir channels (*MacKinnon et al., 1990*; *MacKinnon and Miller, 1989*; *Ramu et al., 2004*). Comparison of the partial sequences of hKir6.2 and mKir6.2 in the region lining the external vestibule of the pore reveals differences at three positions shown in bold (*Figure 4A*, top). To determine which of the three differences underlies the observed differing SpTx1 affinities, we performed a mutagenesis study with

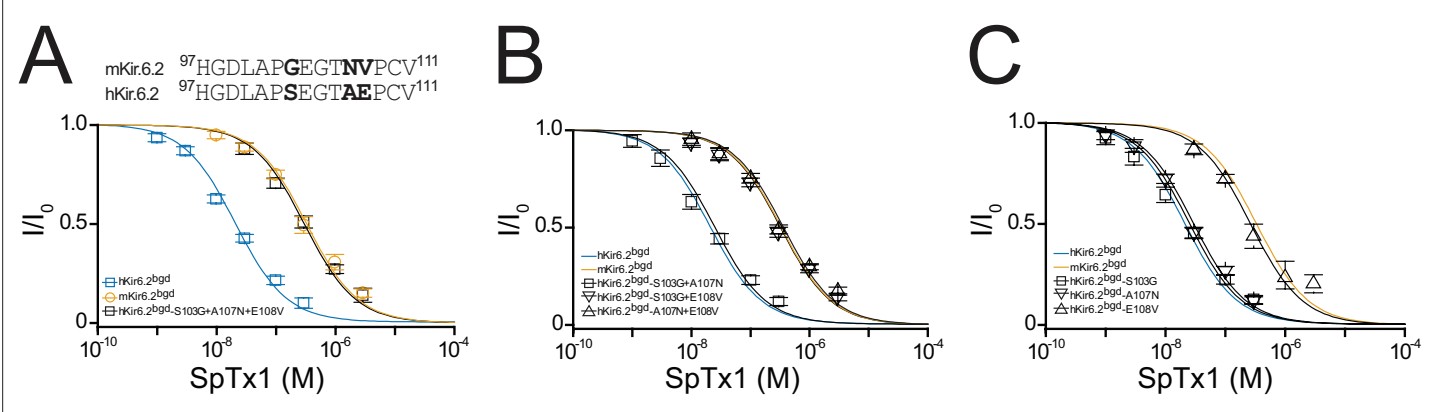

**Figure 4.** Residue E108 underlies the high affinity of hKir6.2 for SpTx1. (**A**) Shown at the top is a comparison of partial amino acid sequences of the extracellular vestibule of the pore between the first and second transmembrane segments in hKir6.2 and mKir6.2. The three residues that differ between the two sequences are bolded. Shown below are fractions of remaining currents of hKir6.2$^{bgd}$ and mKir6.2$^{bgd}$ plotted against the concentration of SpTx1 along with those of hKir6.2$^{bgd}$ containing an additional triple mutation (S103G + A107N + E108V) to mimic mKir6.2$^{bgd}$. The curves superimposed on data correspond to the fits of an equation for a bimolecular reaction. The fitted $K_d$ values are 2.08 (±0.18) × 10$^{-8}$ M for hKir6.2$^{bgd}$ (blue squares), 3.33 (±0.43) × 10$^{-7}$ M for mKir6.2$^{bgd}$ (orange circles), and 3.11 (±0.34) × 10$^{-7}$ M for hKir6.2$^{bgd}$ with the triple mutation (black squares). Fractions of remaining currents of hKir6.2$^{bgd}$ containing additional individual double (**B**) or single (**C**) mutations. The fitted $K_d$ values are 2.48 (±0.28) × 10$^{-8}$ M for hKir6.2$^{bgd}$ containing the double mutations S103G and A107N (squares), 3.23 (±0.39) × 10$^{-7}$ M for hKir6.2$^{bgd}$ containing S103G and E108V (inverse triangles), and 3.63 (±0.44) × 10$^{-7}$ M for hKir6.2$^{bgd}$ containing A107N and E108V (triangles) in (**B**), or 2.51 (±0.30) × 10$^{-8}$ M for hKir6.2$^{bgd}$ containing the single mutation S103G (squares), 2.93 (±0.28) × 10$^{-8}$ M for hKir6.2$^{bgd}$ containing A107N (inverse triangles), and 2.64 (±0.34) × 10$^{-7}$ M for hKir6.2$^{bgd}$ containing E108V (triangles) in (**C**). For ease of comparison, the fitted curves (blue and orange) for hKir6.2$^{bgd}$ and mKir6.2$^{bgd}$ from (**A**) are also replotted in (**B, C**). All data are plotted as mean ± standard error of the mean (SEM; $n$ = 5 independent experiments).

The online version of this article includes the following source data for figure 4:

**Source data 1.** Related to *Figure 4A–C*.

hKir6.2$^{bgd}$ first. The triple mutations (S103G, A107N, and E108V), making this segment identical to that in mKir6.2, lowered the affinity of hKir6.2$^{bgd}$ for SpTx1 to the level of mKir6.2$^{bgd}$ (*Figure 4A*, black squares).

We then mutated two of the three residues at a time. The SpTx1 affinity of the mutant S103G + A107N, which did not carry E108V, remained as high as that of hKir6.2$^{bgd}$ itself (*Figure 4B*, squares versus blue curve). In contrast, the remaining two mutants carrying the mutation E108V exhibited markedly reduced SpTx1 affinities similar to that of mKir6.2$^{bgd}$ (*Figure 4B*, triangles and inverse triangles versus orange curve). These mutagenesis results point to E108V as the mutation responsible for lowering the affinity of hKir6.2$^{bgd}$ for SpTx1. To further confirm this inference, we examined the effects of individual point mutations. Indeed, the E108V mutation alone lowered the affinity of hKir6.2$^{bgd}$ to about that of mKir6.2$^{bgd}$ (*Figure 4C*, triangles versus orange curve) whereas the other two point mutations had little effects on the affinity of hKir6.2$^{bgd}$ (*Figure 4C*, squares and inverse triangles versus blue curve). We also performed the mutagenesis on mKir6.2$^{bgd}$, mKir6.2 + mSUR1 and hKir6.2 + hSUR1 (*Figure 5*). Either the reversed triple or single-V108E mutation conferred such a high SpTx1 affinity on mKir6.2$^{bgd}$ that was comparable to the SpTx1 affinity of hKir6.2$^{bgd}$ (*Figure 5A–C, F*, triangles and circles versus blue curve).

To demonstrate that the above results also occur in the octameric K$_{ATP}$ channel–protein complex, we examined hKir6.2 with the E108V mutation and mKir6.2 with the V108E mutation coexpressed with their respective SUR1. Indeed, the m*Kir6.2$^{V108E}$* (mKir6.2-V108E) mutation conferred the high SpTx1 affinity of hK$_{ATP}$ channels on mK$_{ATP}$ channels coexpressed with mSUR1 (*Figure 5D, G*, inverse triangles versus cyan curve). Conversely, the hKir6.2-E108V mutation conferred the low SpTx1 affinity of mK$_{ATP}$ channels on hK$_{ATP}$ channels coexpressed with hSUR1 (*Figure 5E, G*, squares versus vermilion curve).

The above results demonstrate that the low affinity of mK$_{ATP}$ channels for SpTx1 reflects a residue variant, valine versus glutamate, at amino acid position 108 located in the extracellular vestibule of the pore. This finding in turn strengthens the notion that SpTx1 inhibits the K$_{ATP}$ channel by blocking Kir6.2's ion-conduction pore (*Ramu et al., 2018*; *Ramu and Lu, 2019*).

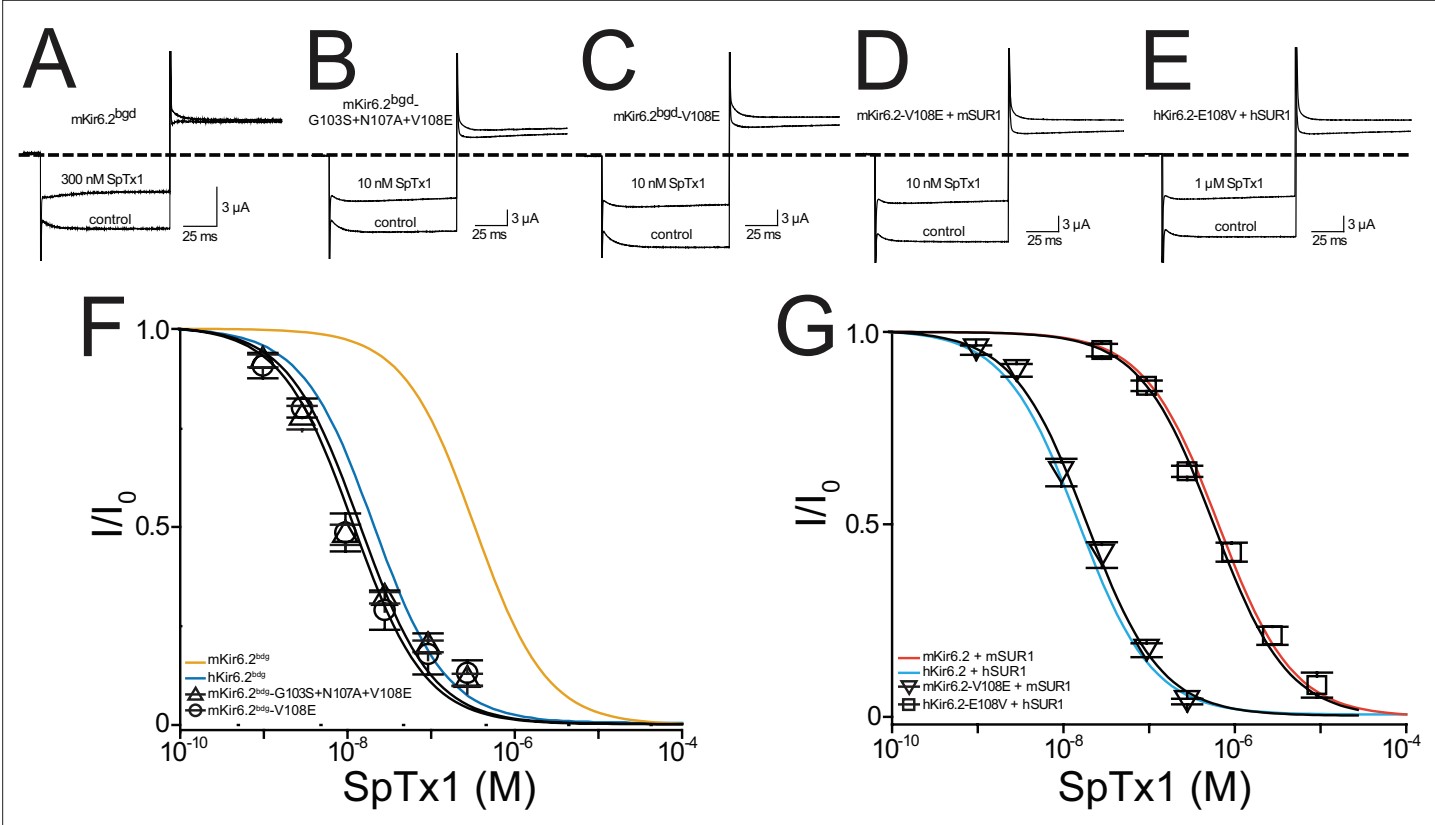

**Figure 5.** A single-point mutation confers high SpTx1 affinity on mKir6.2$^{bgd}$ and switches the affinities of hK$_{ATP}$ and mK$_{ATP}$ channels for SpTx1. Currents of mKir6.2$^{bgd}$ (**A**), mKir6.2$^{bgd}$ containing the triple mutations G103S, N107A, and V108E (**B**), mKir6.2$^{bgd}$ containing the single mutation V108E (**C**), azide-activated mKir6.2-V108E coexpressed with mSUR1 (**D**), or azide-activated hKir6.2-E108V with hSUR1 (**E**). The currents were elicited by stepping voltages from the holding potential of 0 to −80 mV and then to +80 mV in the presence of 100 mM K$^+$ in the bath solution, and recorded in the absence (control) or presence of SpTx1 as indicated. The dashed line indicates zero-current level. (**F, G**) Fractions of remaining channel currents plotted against the concentration of SpTx1. The curves superimposed on data correspond to the fits of an equation for a bimolecular reaction. The fitted $K_d$ values are 1.42 (±0.25) × 10$^{-8}$ M for mKir6.2$^{bgd}$ containing the triple mutations (triangles) and 1.20 (±0.19) × 10$^{-8}$ M for mKir6.2$^{bgd}$ containing V108E (circles) in (**F**), or 1.99 (±0.19) × 10$^{-8}$ M for mKir6.2-V108E coexpressed with mSUR1 (inverse triangles) and 6.16 (±0.48) × 10$^{-7}$ M for hKir6.2-E108V with hSUR1 (squares) in (**G**), where data are plotted as mean ± standard error of the mean (SEM; $n$ = 5 independent experiments). For ease of comparison, the fitted curves (blue and orange) for hKir6.2$^{bgd}$ and mKir6.2$^{bgd}$ in (**F**) are replotted from *Figure 4A*, and those curves (cyan and vermilion) for hKir6.2 with hSUR1 and mKir6.2 with mSUR1 in (**G**) are replotted from *Figure 3E*.

The online version of this article includes the following source data for figure 5:

**Source data 1.** Related to *Figure 5A–G*.

## SpTx1 depolarizes the membrane potential of β cells in pancreatic islets from V108E-mutant mice

On the basis of the mechanistic information revealed by the above heterologous mutagenesis studies, we generated a SpTx1-sensitive-mutant mouse line, in which the valine 108 residue of the endogenous (Endo) mKir6.2 was replaced by a glutamate residue, dubbed $^{Endo}$mKir6.2$^{V108E}$. According to the prevailing paradigm, if the high SpTx1 sensitivity is conferred on the K$_{ATP}$ channels of $^{Endo}$mKir6.2$^{V108E}$ mice, then SpTx1 should depolarize $V_m$ of their β cells and possibly elicit action potentials.

As performed with wild-type islet cells (*Figure 2*), we recorded $V_m$ from individual β cells in isolated but intact $^{Endo}$mKir6.2$^{V108E}$ islets (*Figure 6*, *Figure 6—figure supplement 1*). Starting from 0, 5, or 8 mM glucose in the extracellular medium, the islets were subsequently challenged with high glucose, SpTx1, or glibenclamide. In a nominal glucose-free condition, SpTx1 at 0.2 µM, 10 times higher than the $K_d$ value for V108E-carrying mKir6.2 (+mSUR1), caused a small depolarization (*Figure 6A*, orange segment); this SpTx1-induced $V_m$ depolarization was, however, insufficient to elicit action potentials in comparison to that induced by high glucose challenge or glibenclamide that elicited action potentials

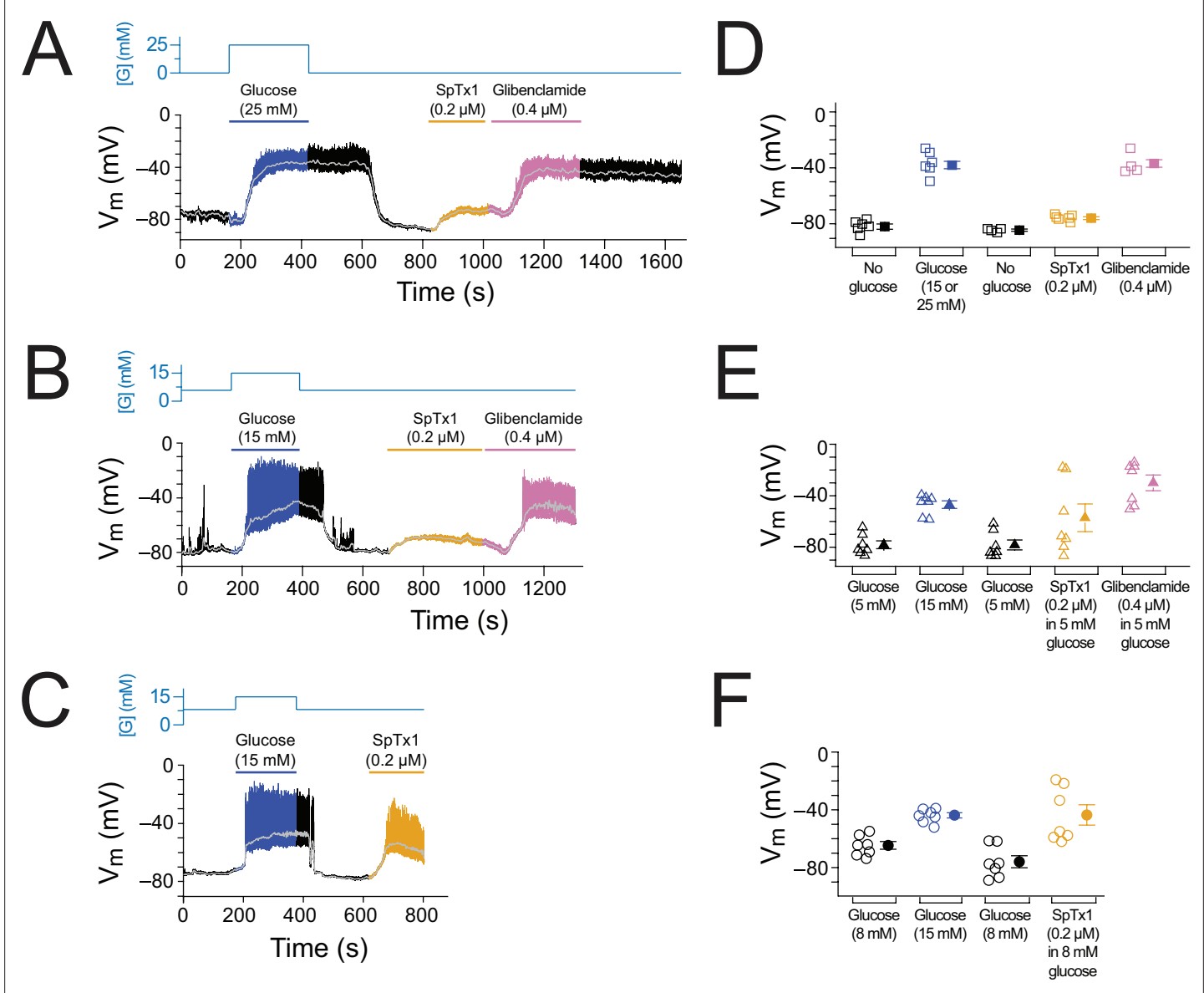

**Figure 6.** SpTx1 depolarizes the membrane potential ($V_m$) of β cells in isolated pancreatic islets from $^{Endo}$mKir6.2$^{V108E}$ mice. (**A–C**) $V_m$ traces recorded in the perforated whole-cell mode from individual β cells near the surface of isolated intact islets from $^{Endo}$mKir6.2$^{V108E}$ mice (8–12 weeks of age). The switching of the glucose concentration from 0 mM (**A**), 5 mM (**B**), or 8 mM (**C**) to 15 or 25 mM and back is indicated by the blue schematic line at the top, and the application of 0.2 μM SpTx1 (orange) or 0.4 μM glibenclamide (purple) or the presence of 15 or 25 mM glucose in the bath solution is as indicated by their color-coded lines above the $V_m$ trace. The light gray curve overlaid on the $V_m$ trace was obtained by filtering the recorded trace at 0.1 Hz using a low-pass Gaussian routine. (**D–F**) Dot plots of filtered $V_m$ values for individual cells from multiple islets under the conditions of the corresponding illustrative traces shown in (**A–C**), where their mean ± standard error of the mean (SEM) are plotted on the right as filled symbols with errors bars. Squares, $n$ = 4 or 6 independent experiments, are as plotted in (**D**); triangles, $n$ = 7 in (**E**); circles, $n$ = 7 in (**F**). Under the 5 mM glucose condition, SpTx1 had variable effect on $V_m$ of β cells in multiple islets (**E**), but the toxin did not elicit any action potentials regardless of the magnitude of apparent depolarization.

The online version of this article includes the following source data and figure supplement(s) for figure 6:

**Source data 1.** Related to *Figure 6A–F*.

**Figure supplement 1.** DNA sequencing result and analysis of $^{Endo}$mKir6.2$^{V108E}$ mice.

**Figure supplement 1—source data 1.** DNA sequencing chromatogram containing the proximal region of the endogenous Kir6.2 gene in $^{Endo}$mKir6.2$^{V108E}$ mice related to *Figure 6—figure supplement 1A*.

**Figure supplement 1—source data 2.** DNA sequencing chromatogram containing the distal region of the endogenous Kir6.2 gene in $^{Endo}$mKir6.2$^{V108E}$ mice related to *Figure 6—figure supplement 1A*.

(*Figure 6A*, blue and pink segments). With 5 mM glucose, SpTx1 at 0.2 μM caused (variable) $V_m$ depolarization without eliciting any action potentials, which was again less than that induced by high glucose challenge or glibenclamide (*Figure 6B*). However, with 8 mM glucose, SpTx1 at 0.2 μM strongly depolarized $V_m$ and elicited action potentials (*Figure 6C*). The results pooled from multiple measurements under all above three glucose conditions are summarized in *Figure 6D–F*. Thus, the impact of SpTx1 on $V_m$ depends on the glucose concentration.

## SpTx1 markedly potentiates GSIS from pancreatic islets of V108E-mutant mice

We examined GSIS of islets isolated from the pancreas of $^{Endo}$mKir6.2$^{V108E}$ mice by a perifusion method. Given that SpTx1 in 0 and 5 mM glucose conditions did not trigger action potentials but it did so in the presence of 8 mM glucose, we set the basal glucose level to 3 mM, a known nonstimulating concentration for isolated intact islets, and then raised the glucose concentration to 10 mM to stimulate insulin secretion in the initial set of studies. As a positive control, upon elevating the glucose concentration in the perifusion solution from 3 to 10 mM, insulin secretion rapidly increased in the first phase and then declined in the second phase to a relatively steady level above the baseline (*Figure 7A*, black trace). After the glucose concentration was lowered back to 3 mM, insulin secretion returned to the baseline level. As an additional control, elevation of glucose and inclusion of 1 μM glibenclamide together yielded insulin secretion much greater than that from elevated glucose alone. As expected, even after removing glibenclamide from the perifusion solution and lowering glucose back to 3 mM, the elevated insulin secretion largely persisted (*Figure 7A*, cyan trace).

Next, we tested whether SpTx1 could act as an effective secondary secretagogue to potentiate GSIS in 10 mM glucose. Because 0.2–1 μM SpTx1 did not potentiate insulin secretion from isolated wild-type mouse islets in 7 or 15 mM glucose (*Figure 1C*), we first tested the effect of a SpTx1 concentration within this range on GSIS from isolated islets of $^{Endo}$mKir6.2$^{V108E}$ mice. Inclusion of 0.5 μM SpTx1 in the perifusion solution potentiated insulin secretion in 10 mM glucose to a level between that in 10 mM glucose alone and in 10 mM glucose plus glibenclamide (*Figure 7A*, orange trace). Following removal of high glucose by subsequent perifusion with a solution containing 3 mM glucose, which also washed out extracellularly bound SpTx1, insulin secretion returned to the baseline level. This potentiating effect of SpTx1 on GSIS from islets of $^{Endo}$mKir6.2$^{V108E}$ mice, not from those of wild-type mice, strongly implies that SpTx1 generated this effect by blocking Kir6.2 channels harboring the introduced V108E mutation.

We wondered whether SpTx1 diffused slowly into islets to reach β cells, and a preincubation of islets with SpTx1 would thus produce a greater potentiation of GSIS. To examine this possibility, we added SpTx1 10 min before raising glucose concentration to allow more time for equilibration. We found that, whether added earlier or at the same time as raising the glucose concentration, SpTx1 had comparable potentiating effects on GSIS in 10 mM glucose (*Figure 7B*). The potentiating effects of glibenclamide were also comparable regardless of whether it was applied concurrently with or before glucose elevation (*Figure 7C*). Adding these two inhibitors 10 min before the high glucose challenge also provided us an opportunity to evaluate SpTx1 and glibenclamide as primary secretagogues, triggering insulin secretion in the presence of a nonstimulating concentration of glucose. In the presence of nonstimulating 3 mM glucose, glibenclamide triggered sizable secretion of insulin (*Figure 7C*, purple trace) whereas SpTx1 caused no detectable insulin secretion above the baseline level (*Figure 7B*, blue trace).

Furthermore, we also examined the effect of SpTx1 on insulin secretion in the presence of 16.7 mM glucose, which is an empirically determined optimal concentration commonly used to stimulate a very high level of insulin secretion from mouse islets (*Alcazar and Buchwald, 2019*; *Cerasi, 1975*; *Malaisse et al., 1967*). Either glibenclamide or SpTx1 was added into the perifusion solution containing 3 mM glucose, 10 min before raising glucose to 16.7 mM. In the presence of 3 mM glucose, glibenclamide again stimulated sizable secretion of insulin, which decreased somewhat with time (*Figure 7D*, purple trace). Raising glucose to 16.7 mM in the presence of glibenclamide caused much greater secretion of insulin than it did in the absence of glibenclamide (*Figure 7D*, purple trace versus black trace). The same procedure was repeated with SpTx1. In the presence of 3 mM glucose, 0.5 μM SpTx1 did not stimulate insulin secretion above the baseline level but did markedly potentiate insulin secretion in the presence of 16.7 mM glucose (*Figure 7D*, blue trace versus black trace), albeit less than that

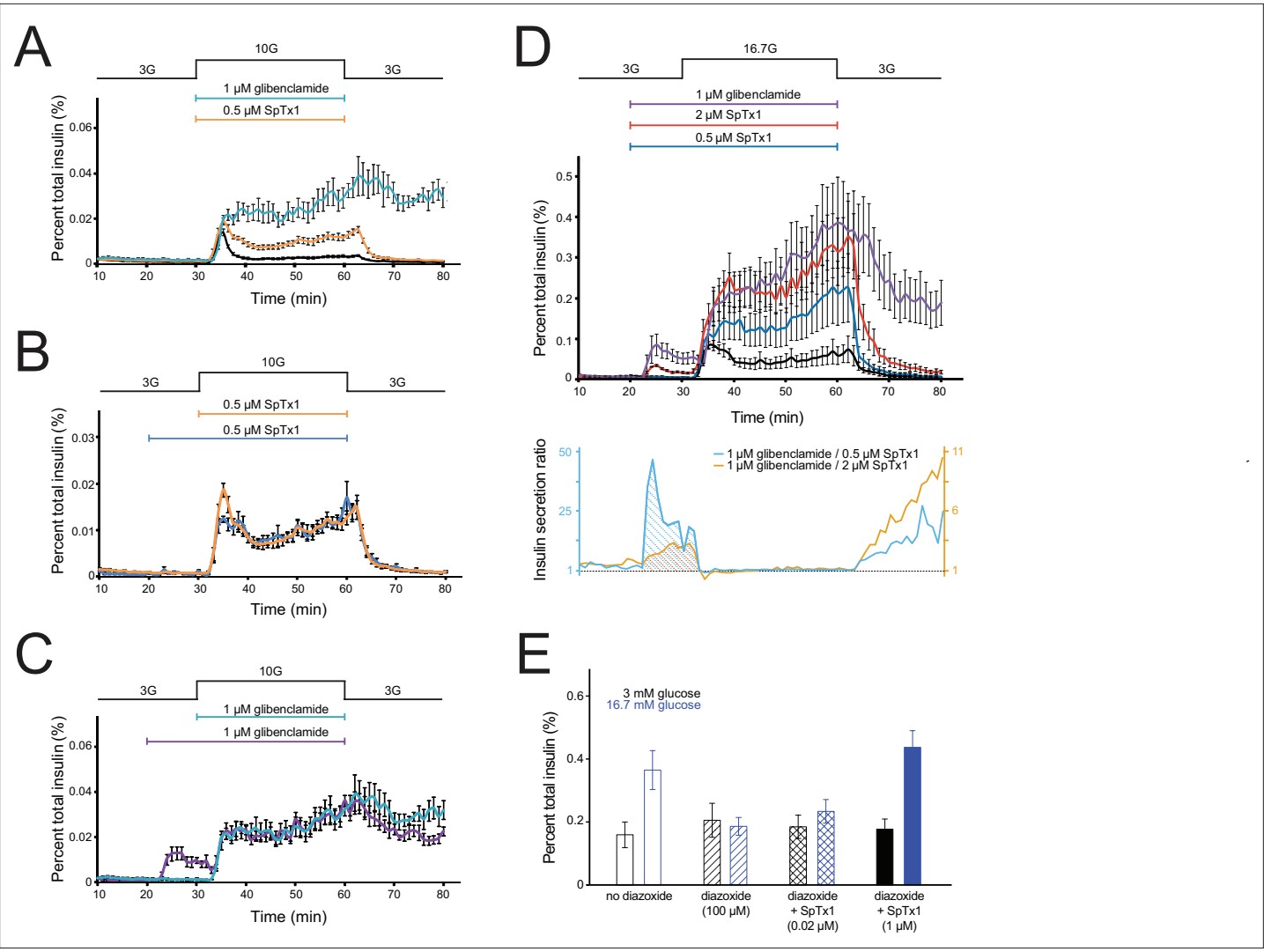

**Figure 7.** SpTx1 potentiates glucose-stimulated insulin secretion (GSIS) from isolated pancreatic islets of [Endo]mKir6.2[V108E] mice and counteracts the effect of diazoxide on GSIS. (**A–D**) The amount of insulin per ml perifusion solution released from ~180 isolated islets of [Endo]mKir6.2[V108E] mice (8–12 weeks of age) as a percentage of the total insulin content is plotted against time. The rate of perifusion was 1 ml/min. All data points are plotted as mean ± SEM connected by lines color coded for specified conditions. The elevation of the glucose concentration in the perifusion solution from 3 to 10 mM (**A–C**) or 16.7 mM (**D**) and the return to 3 mM are as indicated by the black schematic outline at the top. In (**A**), the cyan, orange, or black data plot represents the insulin secretion profile with the inclusion of 1 μM glibenclamide ($n = 4$ independent experiments), 0.5 μM SpTx1 ($n = 6$), or neither ($n = 6$). In (**B**), the blue or orange data plot represents the insulin secretion profile with the inclusion of 0.5 μM of SpTx1 either 10 min prior to ($n = 4$) or at the time of raising the glucose concentration. In (**C**), the purple or cyan data plot represents the insulin secretion profile with the inclusion of 1 μM of glibenclamide either 10 min prior to ($n = 4$) or at the time of raising the glucose concentration. For ease of comparison, the orange data curve in (**B**) and cyan data curve in (**C**) are replotted from (**A**). In (**D**), the purple, vermilion, blue, and black data plots represent the insulin secretion profile with the inclusion of 1 μM glibenclamide ($n = 3$), 2 μM SpTx1 ($n = 3$), 0.5 μM SpTx1 ($n = 3$), or neither inhibitor ($n = 3$). The orange or cyan curve shown below the insulin secretion profiles represents the ratio between the amounts of insulin secreted in the presence of 1 μM glibenclamide and that in the presence of 0.5 μM SpTx1 (cyan) or 2 μM SpTx1 (orange). (**E**) Histograms of GSIS from isolated pancreatic islets of 8- to 12-week-old [Endo]mKir6.2[V108E] mice. For each independent experiment under a specified condition, individual groups of 5 or 10 islets were placed in the wells of a microwell plate for insulin release assay. The secreted insulin as a percentage of the total insulin content of the islets (mean ± standard error of the mean [SEM]; $n = 6$ for each case) is presented as histograms where black or blue color codes for the presence of 3 or 16.7 mM glucose in the bathing medium. The fill pattern inside each pair of rectangles represents a tested condition: glucose only (open), glucose + 100 μM diazoxide (diagonal lines), glucose + 100 μM diazoxide + 0.02 μM SpTx1 (crossed lines), and glucose + 100 μM diazoxide + 1 μM SpTx1 (filled). For the two key group comparisons, the p value of two-tailed Student's *t*-test is 0.034 between the glucose only group and the glucose + 100 μM diazoxide group and is 0.003 between the glucose + 100 μM diazoxide group and the glucose + 100 μM diazoxide + 1 μM SpTx1 group, all in the presence of 16.7 mM glucose.

The online version of this article includes the following source data for figure 7:

**Source data 1.** Related to *Figure 7A–E*.

potentiated by 1 µM glibenclamide (*Figure 7D*, blue trace versus purple trace). As we increased the SpTx1 concentration to 2 µM, SpTx1 potentiated insulin secretion in the presence of 16.7 mM glucose to a level comparable to that caused by 1 µM glibenclamide (*Figure 7D*, vermilion trace versus purple trace in the ~30 to ~60 min region). These observations establish a pair of practically equivalent concentrations for the two inhibitors to act as secondary insulin secretagogues. This is perhaps not surprising because 1 µM glibenclamide in the presence of physiological concentrations of intracellular ATP and 2 µM SpTx1 ($K_d$ = 20 nM; *Figure 5G*) are both expected to inhibit 99% of the $K^+$ current carried by their target channels on the cell surface (*Proks et al., 2013*). However, at these equivalent concentrations, glibenclamide and SpTx1 did not have comparable abilities as primary insulin secret-agogues; SpTx1 stimulated much less insulin secretion than glibenclamide did as evidenced by their differing effects in nonstimulating 3 mM glucose (*Figure 7D*, vermilion trace versus purple trace in the ~20 to ~30 min region). These comparable secondary secretagogue effects but differing primary secretagogue effects between glibenclamide and SpTx1 are more clearly shown by the ratio of the mean insulin secretion in the presence of 1 µM glibenclamide to that in the presence of 2 µM SpTx1 (*Figure 7D*, shaded area under the orange curve in the lower panel). Such a difference in the ratio was even more pronounced when 1 µM glibenclamide and 0.5 µM SpTx1 were compared (*Figure 7D*, shaded area under the cyan curve in the lower panel).

## SpTx1 counteracts the antagonistic effect of diazoxide on GSIS from pancreatic islets of V108E-mutant mice

If SpTx1 acts as an effective secondary insulin secretagogue by blocking the V108E-mutant mKir6.2, then SpTx1 should counteract the effect of diazoxide, an opener of $K_{ATP}$ channels. In a static assay, 16.7 mM glucose stimulated insulin secretion from isolated $^{Endo}$mKir6.2$^{V108E}$ mouse islets to a level above the baseline level in 3 mM glucose (*Figure 7E*). Furthermore, 100 µM diazoxide lowered the insulin secretion in 16.7 mM glucose to the baseline level. SpTx1 at 1 µM, but not 0.02 µM, could effectively boost GSIS in the presence of diazoxide. As expected, in the presence of nonstimulating 3 mM glucose, the levels of insulin secretion were comparably low under the four compared conditions.

## SpTx1 increases the plasma insulin level and lowers the blood glucose concentration in diabetic mice

Our ability to confer high SpTx1 sensitivity on mKir6.2 in mice enabled us to examine whether SpTx1 can actually trigger impactful insulin secretion and thus lower the elevated blood glucose level in diabetic mice that overexpress constitutively active mutant mKir6.2 (*Figure 8*). We started with an NDM-model diabetic-mouse line previously constructed by the Nichols' group, which contained an inducible mutant mKir6.2 transgene in the *Rosa26* locus (*Pdx$^{CreERT2}$*; *Rosa26$^{lsl-Kcnj11}$*), hereafter referred to as $^{Rosa26}$mKir6.2$^{NDM}$ (*Remedi et al., 2009*). After a 9-day induction to overexpress the constitutively active mutant mKir6.2 in the pancreas of $^{Rosa26}$mKir6.2$^{NDM}$ mice, the overnight-fasted blood glucose level of mice remained highly elevated during the 2-hr observation period (*Figure 8A*). We found that an IV injection of SpTx1 (1 mg/kg) or its vehicle saline had no meaningful effect on the elevated blood glucose levels of diabetic $^{Rosa26}$mKir6.2$^{NDM}$ mice and did not trigger a rise of insulin in the plasma of their circulating blood collected during the observation period (*Figure 8A, C*, filled and open orange triangles). These findings are expected because mKir6.2 exhibits very low affinity for SpTx1.

To increase the affinity of mKir6.2 in $^{Rosa26}$mKir6.2$^{NDM}$ mice for SpTx1, we introduced the V108E mutation into all endogenous and transgenic copies of mKir6.2 within their genome (*Figure 8—figure supplement 1*, *Figure 8—figure supplement 2*). The resulting mice are denoted here as $^{Rosa26}$mKir6.2$^{NDM-V108E}$. Like the original $^{Rosa26}$mKir6.2$^{NDM}$ mice, after a 9-day induction to overexpress the mutant mKir6.2 in the pancreas of $^{Rosa26}$mKir6.2$^{NDM-V108E}$ mice, their overnight-fasted blood glucose level also became highly elevated in a sustained manner. A single bolus application of glibenclamide (40 mg/kg) or its vehicle DMSO neither markedly lowered the elevated blood glucose level nor caused a rise of insulin in the plasma (*Figure 8A, C*, filled and open blue circles) of diabetic $^{Rosa26}$mKir6.2$^{NDM-V108E}$ mice during the observation period. This finding is consistent with the ineffectiveness of acute glibenclamide treatments reported in the original study of $^{Rosa26}$mKir6.2$^{NDM}$ mice (*Remedi et al., 2009*). Thus, in the present context, $^{Rosa26}$mKir6.2$^{NDM-V108E}$ mice retained the basic characteristics of inducible diabetic $^{Rosa26}$mKir6.2$^{NDM}$ mice.

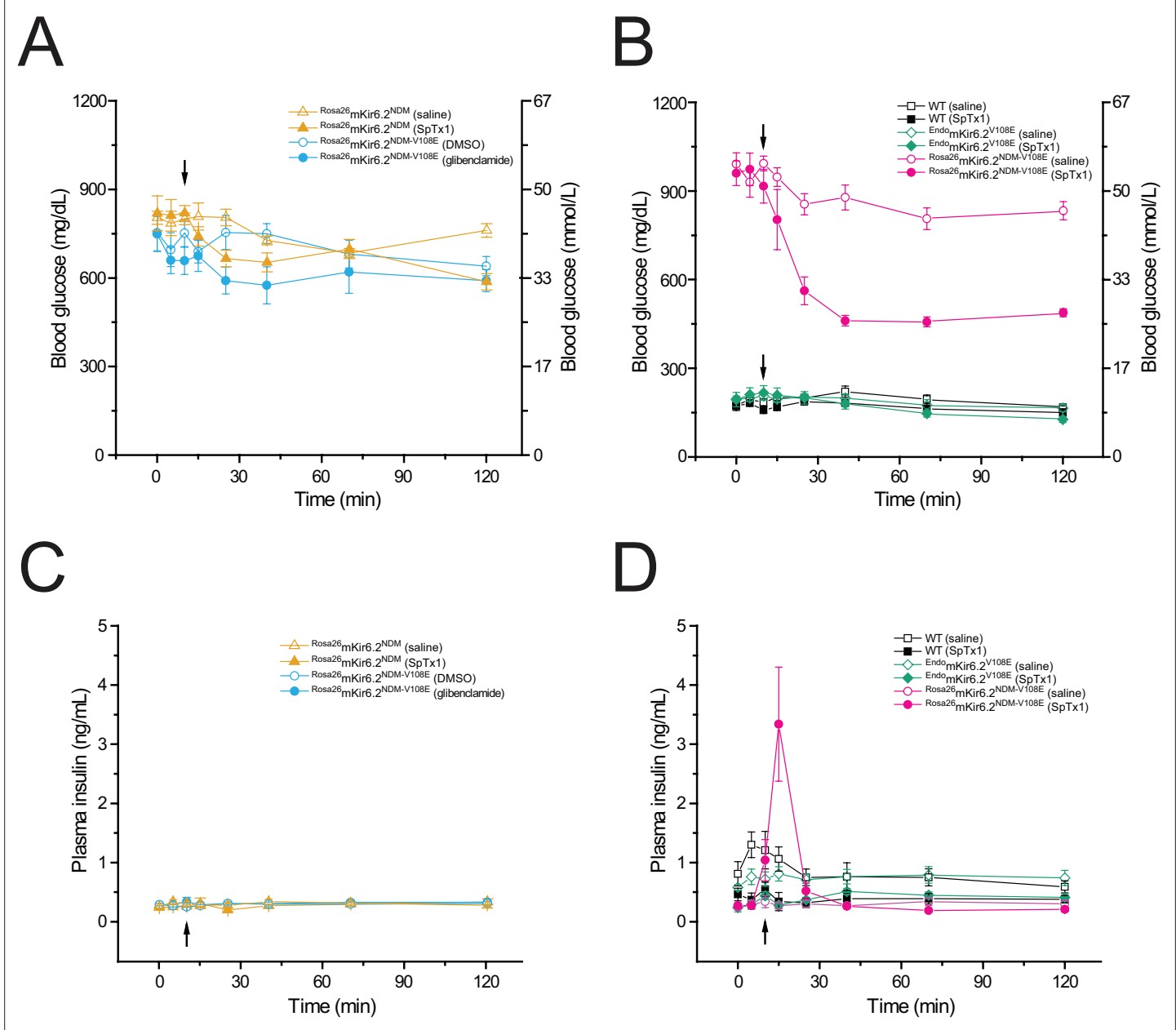

**Figure 8.** SpTx1 triggers a transient rise of plasma insulin and lowers the elevated blood glucose level in diabetic [Rosa26]mKir6.2[NDM-V108E] mice. Blood glucose (**A, B**) and corresponding plasma insulin (**C, D**) levels (mean ± standard error of the mean [SEM]) of overnight-fasted mice (8–12 weeks of age) at indicated time points during a 2-hr observation period. (**A, C**) SpTx1 (1 mg/kg, filled orange triangles) or its vehicle saline (open orange triangles) was intravenously administered in [Rosa26]mKir6.2[NDM] mice (n = 5 independent experiments for each case) and glibenclamide (40 mg/kg, filled cyan circles) or its vehicle DMSO (open cyan circles) was intraperitoneally administered in [Rosa26]mKir6.2[NDM-V108E] mice (n = 5 each) as indicated by the arrow. (**B, D**) SpTx1 (1 mg/kg, filled symbols) or its vehicle saline (open symbols) was intravenously administered in wild-type (black squares, n = 5 each), [Endo]mKir6.2[V108E] (green diamonds, n = 5 each), and [Rosa26]mKir6.2[NDM-V108E] (magenta circles, n = 10 each) mice as indicated by the arrow. For the comparison of the plasma insulin levels of [Rosa26]mKir6.2[NDM-V108E] mice at 5-min post administration, the p value of two-tailed Student's t-test is 0.011 between the vehicle group and the SpTx1 group.

The online version of this article includes the following source data and figure supplement(s) for figure 8:

**Source data 1.** Related to *Figure 8A–D*.

**Figure supplement 1.** DNA sequencing results and analysis of [Rosa26]mKir6.2[NDM-V108E] mice.

**Figure supplement 1—source data 1.** DNA sequencing chromatogram containing the proximal region of the endogenous Kir6.2 gene in [Rosa26]mKir6.2[NDM-V108E] mice related to *Figure 8—figure supplement 1A*.

*Figure 8 continued on next page*

Figure 8 continued

**Figure supplement 1—source data 2.** DNA sequencing chromatogram containing the distal region of the endogenous Kir6.2 gene in $^{Rosa26}$mKir6.2$^{NDM-V108E}$ mice related to *Figure 8—figure supplement 1A*.

**Figure supplement 1—source data 3.** DNA sequencing chromatogram containing the proximal region of the NDM-causing mutant Kir6.2 transgene in $^{Rosa26}$mKir6.2$^{NDM-V108E}$ mice related to *Figure 8—figure supplement 1B*.

**Figure supplement 1—source data 4.** DNA sequencing chromatogram containing the mid region of the NDM-causing mutant Kir6.2 transgene in $^{Rosa26}$mKir6.2$^{NDM-V108E}$ mice related to *Figure 8—figure supplement 1B*.

**Figure supplement 1—source data 5.** DNA sequencing chromatogram containing the distal region of the NDM-causing mutant Kir6.2 transgene in $^{Rosa26}$mKir6.2$^{NDM-V108E}$ mice related to *Figure 8—figure supplement 1B*.

**Figure supplement 2.** The N107Y mutation does not reduce the SpTx1 sensitivity of hKir6.2$^{bgd}$.

**Figure supplement 2—source data 1.** The N107Y mutation does not alter SpTx1 sensitivity of hKir6.2 related to *Figure 8—figure supplement 2*.

Next, we tested whether SpTx1 could mitigate hyperglycemia of diabetic $^{Rosa26}$mKir6.2$^{NDM-V108E}$ mice. Following an IV injection of SpTx1 (1 mg/kg) but not of saline, the elevated blood glucose level of diabetic $^{Rosa26}$mKir6.2$^{NDM-V108E}$ mice dropped markedly over 30 min and this lower level of blood glucose persisted over the remaining observation period (*Figure 8B*, filled and open magenta circles). In contrast, administered SpTx1 or saline had no discernible effects on the fasting blood glucose levels of wild-type and $^{Endo}$mKir6.2$^{V108E}$ mice (*Figure 8B*, filled black squares and green diamonds or open black squares and green diamonds). Moreover, SpTx1, but not saline, triggered a transient but sizable rise of insulin in the plasma of diabetic $^{Rosa26}$mKir6.2$^{NDM-V108E}$ mice, but not in that of wild-type or $^{Endo}$mKir6.2$^{V108E}$ mice (*Figure 8D*). These results show that SpTx1 promotes transient insulin secretion and markedly lowers the highly elevated blood glucose level of diabetic $^{Rosa26}$mKir6.2$^{NDM-V108E}$ mice.

## Discussion

We originally discovered SpTx1 on the basis of its potent inhibition of hKir6.2 with a $K_d$ of 15 nM. In the present study, we found that at 0.2 µM concentration, SpTx1 failed to depolarize β cells, and at a concentration as high as 1 µM, SpTx1 had no discernable effects on GSIS from β cells in isolated pancreatic islets of wild-type mice (*Figures 1C and 2*). This ineffectiveness led us to determine that SpTx1 blocks mKir6.2 with much lower potency than it blocks hKir6.2 (*Figure 3*). Through a series of mutagenesis studies, we found that this low-affinity interaction between mKir6.2 and SpTx1 stems from the presence of a valine residue at position 108 in mKir6.2 whereas it is a glutamate residue in hKir6.2 (*Figure 4*). Replacing the valine residue in mKir6.2 by a glutamate residue confers the high affinity of hKir6.2 for SpTx1 on mKir6.2 (*Figure 5*). This mechanistic information led us to create the SpTx1-sensitive $^{Endo}$mKir6.2$^{V108E}$ mouse line. Using isolated islets of these SpTx1-senstive-mutant mice, we found that contrary to the expectation of Kir6.2 inhibition causing sufficient $V_m$ depolarization and robust insulin secretion, SpTx1 does not impactfully depolarize β cells in glucose concentrations less than 5 mM and it is not an effective primary insulin secretagogue in 3 mM glucose. Nonetheless, SpTx1 depolarizes β cells and elicits action potentials in higher concentrations of glucose and acts as an effective secondary secretagogue, strongly potentiating GSIS from these islets (*Figures 6 and 7*).

SpTx1 inhibits $K_{ATP}$ channels by blocking their ion-conduction pore whereas glibenclamide inhibits the channels by disrupting their nucleotide-dependent gating. Despite this difference, if both SpTx1 and glibenclamide promote insulin secretion solely by inhibiting $K^+$ currents through mKir6.2, increasing the concentration of SpTx1 to a sufficiently high level should produce effects that would match those produced by glibenclamide, in the presence of whether a nonstimulating or an optimal stimulating concentration of glucose. We observed that glibenclamide triggered sizable secretion of insulin in 3 mM glucose and potentiated insulin secretion in 16.7 mM glucose to a much higher level than glucose alone (*Figure 7D*). In contrast, 0.5 µM SpTx1 did not trigger observable insulin secretion in 3 mM glucose or potentiate insulin secretion in 16.7 mM glucose as strongly as glibenclamide. However, at 2 µM, SpTx1 did produce such a strong secondary secretagogue effect in 16.7 mM glucose that it potentiated GSIS to a level comparable to what was caused by glibenclamide, but it failed to act as a strong primary secretagogue to stimulate as much insulin secretion in 3 mM glucose as glibenclamide did. Thus, at a sufficiently high concentration, SpTx1 can act as an

effective secondary insulin secretagogue like glibenclamide but it does not appear to act as an effective primary insulin secretagogue.

One possible reason for these differing effects of SpTx1 and glibenclamide in a nonstimulating glucose concentration versus an optimal stimulating glucose concentration is that SpTx1 potentiated GSIS by acting on an off-target. However, we did not observe any meaningful potentiating effects of SpTx1 on GSIS from isolated islets of wild-type mice (*Figure 1C*, see also *Figure 2*) before we conferred a high SpTx1 affinity on their endogenous mKir6.2 (*Figures 6 and 7*). Furthermore, SpTx1 can counteract the inhibitory effect of the $K_{ATP}$ opener diazoxide on GSIS (*Figure 7E*). Thus, a potential off-target mechanism is unlikely.

An alternative possibility is that SpTx1 does not have all of the pharmacological actions of glibenclamide because glibenclamide may stimulate insulin secretion not solely by inhibiting Kir6.2 channels in the plasma membrane of β cells. Glibenclamide has been shown to cause an additional amount of insulin secretion, even after the membrane potential was clamped to a depolarized level by raising the extracellular $K^+$ to a constant 30 mM such that the impact of any alteration of the Kir6.2 activity on the membrane potential of the β cells and consequently the $Ca_v$ activity would become negligible (*Geng et al., 2007*). Thus, glibenclamide may have one or more additional effective targets other than the cell surface $K_{ATP}$ channels, and hence the extracellular Kir6.2-blocker SpTx1 cannot effectively mimic every action of glibenclamide. Indeed, many studies have shown that glibenclamide interacts with other proteins including syntaxin-1A and Epac2 (*Eliasson et al., 1996*; *Hinke, 2009*; *Kang et al., 2011*; *Lehtihet et al., 2003*; *Renström et al., 2002*; *Shibasaki et al., 2014*; *Tian et al., 1998*; *Zhang et al., 2009*). In principle, glibenclamide may also act on some intracellular $K_{ATP}$ channels that SpTx1 cannot access (*Geng et al., 2003*).

Regarding the application of SpTx1 in in vivo studies, on one hand, SpTx1's characteristics, which allow the toxin to function only as an effective secondary but not as an effective primary secretagogue, make it unsuitable when a strong primary secretagogue effect is needed. On the other hand, being an effective secondary but not an effective primary secretagogue, SpTx1 will only strongly promote insulin secretion when blood glucose level becomes highly elevated but it will not meaningfully stimulate insulin secretion at a resting level of blood glucose – a desirable outcome unlikely to cause unwanted hyperinsulinemia and hypoglycemia (*Asplund et al., 1983*). Consistent with these predictions, we observed that administration of SpTx1 did not trigger insulin secretion or lower the blood glucose levels in <sup>Endo</sup>mKir6.2<sup>V108E</sup> mice, but it caused transient insulin secretion and markedly lowered the highly elevated glucose level in diabetic <sup>Rosa26</sup>mKir6.2<sup>NDM-V108E</sup> mice (*Figure 8*). In the future, it will be interesting to learn in animal models whether the transient characteristic of insulin secretion caused by SpTx1 helps to mitigate problems associated with overstimulation of insulin secretion and 'exhaustion' of β cells, which can occur with the use of sulfonylureas (*Maedler et al., 2005*; *Matthews et al., 1998*).

As experimental tools, sulfonylureas and SpTx1, while both inhibiting $K_{ATP}$ channels, have important and different characteristics. The pancreatic $K_{ATP}$ channels made of Kir6.2 and SUR1 are inhibited by ATP and activated by MgADP (*Gribble and Reimann, 2003*; *Proks et al., 2013*). The stimulatory effect of MgADP is antagonized by binding of sulfonylureas to SUR1. This apparent inhibitory effect of sulfonylureas, together with the ATP-mediated inhibition, decreases the channel open probability ($P_o$) to near zero. However, without ATP, sulfonylureas alone can only inhibit about 2/3 of the channel current. Thus, the inhibitory effect of sulfonylureas depends on the cellular metabolic state. In contrast, the inhibitory effect of SpTx1 on the Kir6.2 pore in the $K_{ATP}$ channel complex is not expected to depend on the cellular metabolic state. This expectation stems from the observation that SpTx1 blocks, comparably well, currents through a constitutively active mutant Kir6.2 and those through $K_{ATP}$ channels activated with azide that lowers the concentration of intracellular ATP (*Figures 3–5*). However, SpTx1 is of little use when the Kir6.2 pore in the $K_{ATP}$ protein complex is inherently insensitive or is experimentally rendered insensitive to SpTx1, such as that of the wild-type mice and those carrying natural or engineered SpTx1-resistant mutations. Sulfonylureas also have limitations in their inhibition of $K_{ATP}$ channels. Mutations in SUR1 could disrupt the nucleotide-dependent gating of $K_{ATP}$ channels such that a substantial amount of $K_{ATP}$ current would remain even in the presence of a near saturating concentration of sulfonylureas (*Proks et al., 2004*). The $K_{ATP}$ complexes containing SUR2 have apparently low sulfonylurea sensitivity, for example, in cardiac and some neuronal $K_{ATP}$ channels (*Nichols, 2016*; *Seino and Miki, 2003*). Another important difference between the two inhibitors is

that SpTx1 is not membrane permeable and acts from the extracellular side whereas sulfonylureas are membrane permeable and lodge into the transmembrane segments of SUR proteins (*Lee et al., 2017*; *Li et al., 2017*; *Martin et al., 2017*; *Schatz et al., 1977*; *Wu et al., 2018*).

Because SpTx1 and sulfonylureas have different characteristics, these inhibitors may have different therapeutic potentials. For example, sulfonylureas have only limited effects on the neurological symptoms of patients with DEND (*Hattersley and Ashcroft, 2005*; *Pipatpolkai et al., 2020*). To this end, it will be interesting to test whether SpTx1 can help to mitigate these neurological disorders in animal models by directly delivering it into the cerebrospinal fluid.

In summary, we have conferred high-affinity SpTx1 inhibition on mKir6.2 using the point-mutation V108E to mimic the human ortholog in a heterologous expression system, and introduced this mutation into mKir6.2 in wild-type and NDM-model mice via genome editing. By studying membrane potential and GSIS of β cells in isolated pancreatic islets of SpTx1-sensitive-mutant $^{Endo}$mKir6.2$^{V108E}$ mice, we have found that SpTx1 inhibition of Kir6.2 channels causes $V_m$ depolarization, eliciting action potentials only in the presence of sufficiently high glucose, and that like the sulfonylurea glibenclamide, it acts as an effective secondary secretagogue to potentiate GSIS. However, unlike glibenclamide, in a low glucose condition SpTx1 does not trigger action potentials or act as an effective primary insulin secretagogue. Furthermore, an application of SpTx1 in $^{Endo}$mKir6.2$^{V108E}$ mice neither triggers meaningful insulin secretion nor lowers their blood glucose from the resting levels. By contrast, in diabetic $^{Rosa26}$mKir6.2$^{NDM-V108E}$ mice, an application of SpTx1 causes a transient rise of plasma insulin and markedly lowers their highly elevated blood glucose level. These features of the present experimental tool SpTx1 point to the potential therapeutic benefit of a Kir6.2 inhibitor with required pharmacological and pharmaceutical characteristics.

# Materials and methods
## Mutagenesis and electrophysiological recordings

All mutant Kir6.2 cDNAs were produced through PCR-based mutagenesis and confirmed by DNA sequencing. The Kir6.2 and SUR1 cRNAs were synthesized with T7 polymerase using the corresponding linearized cDNAs as templates. Channel currents were recorded from *Xenopus* oocytes, which were injected with cRNA encoding specific wild-type or mutant Kir6.2 channels with or without coinjection with cRNA encoding SUR1, through a two-electrode voltage-clamp amplifier (Oocyte Clamp OC-725C, Warner Instruments Corp.), filtered at 1 kHz, and sampled at 10 kHz using an analog-to-digital converter (Digidata 1322A; MDS Analytical Technologies) interfaced with a personal computer. pClamp8 software (MDS Analytical Technologies) was used for amplifier control and data acquisition. All recordings were performed at room temperature. To activate $K_{ATP}$ (Kir6.2 + SUR1) channels, 3 mM azide was added to the bath solution. The resistance of electrodes filled with 3 M KCl was 0.2–0.4 M$\Omega$. To elicit currents through the channels, the membrane potential of oocytes was stepped from the holding potential of 0 to −80 mV then to +80 mV before returning back to 0 mV. The bath solution contained (in mM): 100 KCl, 0.3 CaCl$_2$, 1.0 MgCl$_2$, and 10 N-2-hydroxyethylpiperazine-N-2-ethane sulfonic acid (HEPES), and bovine serum albumin (BSA, 50 μg/ml), where pH was titrated to 7.6 with KOH. All salts were from Millipore-Sigma. Recombinant SpTx1, produced as previously described (*Ramu et al., 2018*), was added to the bath solution in the concentrations as specified in the relevant figures. The number of measurements was determined on basis of previous studies (*Ramu et al., 2018*; *Ramu and Lu, 2019*). The variations in data values reflected both biological variability and technical errors. No experiments were excluded.

Electrophysiological membrane potential ($V_m$) measurements from individual cells in isolated but intact islets cultured on glass coverslips were performed using the perforated whole-cell patch-clamp method as described (*Yang et al., 2021*). Wax-coated electrodes (G85150T, Warner) were back filled with the intracellular solution containing (in mM): 76 K$_2$SO4, 10 KCl, 10 NaCl, 6 MgCl$_2$, 30 mannitol, 30 HEPES, where pH was titrated to 7.2 at 35°C with *N*-methyl-d-glucamine (NMG). The tip of the pipette was filled with this solution and the remainder was back filled with the same solution plus β-escin (6 or 8 μM). The free Mg$^{2+}$ concentration of this solution, taking the divalent cation-chelating action of SO$_4^{2-}$ into consideration, is estimated to be about 2 mM. The extracellular solution contained (in mM): 135 NaCl, 4 KCl, 2 CaCl$_2$, 2 MgCl$_2$, 10 HEPES, where pH was titrated to 7.4 at 35°C with NMG. Glucose was added to this solution as required and the osmolarity was adjusted to ~300 mOsm with

mannitol. All salts were from Millipore-Sigma. Using the solutions described above, the initial input resistance of the electrode was typically 4–8 MΩ. The electrophysiological recordings were performed under a continuous perfusion condition, 0.3–0.4 ml/min (Instech), at ~35°C (TC-124A and TC-344B Warner). Islets were equilibrated in the recording chamber for 10–15 min before measurements were made. Adequate whole-cell access was achieved typically within 5–10 min. SpTx1 was applied with 0.3% (wt/vol) BSA.

The output of the patch-clamp amplifier (Axopatch 200B, Molecular Devices) was digitized at 2 kHz and stored for later analysis using Igor Pro (v8 or v9, Wavemetrics). The liquid junction potential has been subtracted from the results. To estimate the average $V_m$ values under different conditions, the $V_m$ traces were digitally Gaussian filtered at 0.1 Hz and the filtered data points in the last 50 s of a segment of interest were averaged. The filtered traces are superimposed on the illustrative original traces in the figures where appropriate. The variations in data values reflected both biological variability and technical errors (*Figures 2D–F and 6D–F*). No data, which were successfully recorded over sufficient durations that allowed for adequate examination the membrane potential under the various specified conditions, were excluded.

## Generation of $^{Endo}$mKir6.2$^{V108E}$- and $^{Rosa26}$mKir6.2$^{NDM-V108E}$-mutant mice

To confer the hKir6.2-like, high SpTx1 sensitivity on endogenous mKir6.2 channels in wild-type mice, we employed the CRISPR-Cas9 genome editing technique (*Ran et al., 2013*). The sequences of in vitro transcription template of a guide RNA (gRNA) and the single-stranded oligodeoxynucleotide (ssODN) donor templates are provided below.

Synthesized DNA templates (IDT Technologies) for gRNA targeting the mKir6.2 DNA sequence (Genbank, Accession Number NC_000073.6) also contained a T7 promoter sequence. The sequence of the DNA template for the gRNA from the 5′ end to the 3′ was:

GAAATTAATACGACTCACTATAGGGAGA**GCTTGTGACGCAGGGCACAT***GTTTTAGAGCTAGAAA TAGCAAGTTAAAATAAGGCTAGTCCGTTATCAACTTGAAAAAGTGGCACCGAGTCGGTGCTTTTTT* where the T7 promoter sequences are underlined, target DNA sequences bolded, and tracrRNA italicized; whereas that for the ssODN donor template was:

GGACCTCGATGGAGAAAAGGAAGGCAGATGAAAAGGAGTGGATGCTTGTGACGCAGGGC **T**CGTTAGTGCCCTCTCCGGGGGCCAGGTCACCGTGGGCGAAGGCGATGAGCCACCAGACCAT GGCAAAGA where the sequence of the restriction enzyme (BanII) cleavage site is underlined, the intended nonsynonymous substitution is bolded and synonymous ones italicized (*Figure 6—figure supplement 1* and *Figure 8—figure supplement 1*).

In vitro transcription from the gRNA template or a Cas9 plasmid (T7-Cas9-HA-2NLS) was performed to produce gRNA or Cas9 mRNA using mMESSAGE mMACHINE T7 ULTRA kit (Invitrogen, AM1344). The ssODN stock solution was prepared according to the manufacturer's recommendation (IDT Technologies). The freshly prepared solution, which contained 100 ng/μl Cas9 mRNA, 100 ng/μl gRNA, 200 ng/μl ssODN, 0.1 mM ethylenediaminetetraacetic acid (EDTA), and 10 mM Tris titrated to pH 7.5 with HCl, was injected into cytoplasm of FVB mouse embryos at the Transgenic and Chimeric Mouse Facility (TCMF) of the University of Pennsylvania Perelman School of Medicine (UPENN-PSOM). The injected embryos were implanted into the uterus of pre-prepared surrogate female mice for obtaining germline-transmitting founder $^{Endo}$mKir6.2$^{V108E}$-mutant mice. The intended T323A mutation and the five adjacent nucleotides in the DNA created a cleave site for the restriction enzyme BanII, which was used to assist the identification of genotypes.

Double transgenic $^{Rosa26}$mKir6.2$^{NDM}$ mice on the C57BL/6J (B6J) background were obtained from the laboratory of Dr. Colin Nichols (Washington University, St. Louis, MO; *Remedi et al., 2009*). This mutant mouse line contained two unlinked transgenes. One encodes for a tamoxifen-dependent, pancreas-specific Cre recombinase ($Pdx^{CreERT2}$) and the other is a Cre-inducible copy of NDM-causing mutant Kir6.2 gene inserted at the *Rosa26* locus. The translation of mutant Kir6.2 transgene would harbor the K185Q mutation and lack N-terminal 30 residues (ΔN30). By crossing their $^{Rosa26}$mKir6.2$^{NDM}$ mice with our mutant $^{Endo}$mKir6.2$^{V108E}$ mice, we created a hybrid mutant mouse line on the B6.FVB background whose genome contained the two unlinked transgenes and the endogenous Kir6.2 gene with homozygous V108E mutation. Furthermore, using the hybrid mutant line and the CRISPR-Cas9 targeting strategy described above, we introduced the V108E mutation into the NDM-causing Kir6.2 transgene at the *Rosa26* locus, and the resulting germline-transmitting mutant mice are called

$^{Rosa26}$mKir6.2$^{NDM-V108E}$, in which only the mutant Kir6.2 transgene contain K185Q and ΔN30 mutations but both transgenic and endogenous Kir6.2 genes contain V108E mutation. All mice used in the study were generated and maintained in a UPENN-PSOM animal facility.

## Mice maintenance and their humane endpoints

All mice were housed in ventilated, sterilized polysulfone mouse cages (AN75, Ancare Corp., Bellmore, NY, USA) containing Bed-o'Cobs absorbent bedding (Andersons Lab Bedding, Maumee, OH, USA). Irradiated LabDiet 5053 feed (PicoLab Rodent Diet 20, Fort Worth, TX, USA) and autoclaved acidified water (pH 2.5–2.8 titrated with HCl) were provided ad libitum. Shepherd's shack and nesting material were also provided in the cages to minimize distress of mice. Cages were changed weekly in a Class A2 biological safety cabinet (Baker SG403A-ats, Sanford, ME, USA). The air in the mouse-holding room was exchanged with filtered air at a rate of 15 times of room volume per hour. Room temperature was maintained between 20 and 25°C, and relative humidity between 30% and 70%. Lighting of the room was on and off alternately every 12 hr where the light phase was from 7:00 AM to 7:00 PM. A vivarium-wide rodent health monitoring program showed the absence of the following pathogens: MHV, Sendai virus, MVM, MPV1/2, TMEV, PVM, Reo-3, EDIM, Ectromelia virus, LCMV, MAdV, K-Virus, Polyomavirus, MCMV, Mouse Thymic Virus, Hantavirus, *Mycoplasma pulmonis*, *Citrobacter rodentium*, *Clostridium piliforme*, *Corynebacterioum kutscheri*, CAR *bacillus.*, *Salmonella* sp., *Klebsiella pneumoniae*, *Streptococcus pneumoniae*, *Streptobacillus moniliformis*, *Encephalitozoon cuniculi*, *Myobia musculi*, *Myocoptes musculinus*, *Radfordia affinis*, *Aspiculuris tetraptera*, *Syphacia obvelata*, and *Giardia muris*.

The animal cohort in this study included wild-type and mutant mice at 8–12 weeks of age. The designed endpoint of the study was defined as individual mice reaching that age range and euthanized for various types of experiments. We monitored the general health condition and behavior of each mouse at least three times per week during maintenance and continuously during the 2-hr observation experiment. If a mouse was suspected to have clinical signs of pain and distress, this mouse was then monitored and checked one or several times daily. We used the following criteria to determine the humane endpoints at which mice of any age were euthanized within 1 hr upon our determination: clinical signs of pain and distress, such as hunched posture, inactivity, dehydration, abdominal enlargement caused by intestine obstruction, increased respiratory effort manifested as increased intercostal or subdiaphragmatic retraction and gasping or breathing with an open mouth, raffled fur coat, emaciated body condition (e.g., weight loss of >20%), severe lethargy manifested as unwillingness to ambulate more than a few steps when gently stimulated with a gloved finger, or cold to the touch.

## Mouse genomic DNA isolation and genotyping

For genomic DNA isolation, a tail sample (~2 mm) or a clipped toe from each mouse was digested at 55 °C overnight with gentle mixing in 0.5 ml solution containing 50 mM Tris titrated to pH 8.0 with HCl, 100 mM EDTA, 0.5% sodium dodecyl sulfate (SDS), and 0.5 mg/ml Proteinase K. The high molecular weight genomic DNA in the digestion sample was extracted with a mixture of phenol and chloroform and precipitated using cold ethanol with 0.3 M sodium acetate titrated to pH 6.0. The precipitated genomic DNA was resuspended in 0.1 ml of a solution containing 1 mM EDTA and 10 mM Tris titrated to pH 8.0 with HCl; 1 µl of which was used as the DNA template and added into a 25 µl PCR mixture containing 50 mM KCl, 1.5 mM MgCl$_2$, 20 mM Tris triturated to pH 8.4, 0.2 mM dNTPs, 0.2 µM of each forward and reverse oligodeoxynucleotide primers described below, and 1 U of Platinum Taq DNA polymerase (Invitrogen, 10966034). PCR reactions took place in a thermal cycler (Mastercycler 5333; Eppendorf) with an initial denaturation step at 95°C for 2 min and 30 amplification cycles (denaturation at 95°C for 30 s, annealing at 60°C for 30 s, and extension at 72°C for 45 s for <0.7 kb or 150 s for ~2 kb), and followed by a final 5 min extension step at 72°C. 5 µl of individual PCR products was subjected to electrophoresis on 1% agarose gel, stained with 0.5 µg/ml ethidium bromide, and evaluated against a DNA ladder (Thermo Scientific, SM1331). The initial screen was performed on all resulting PCR products by the expected size (~2 kb) and by their susceptibility to BanII restriction enzyme digestion.

By subsequent DNA sequencing of the whole ~2 kb PCR product, we verified that the BanII-cut products contained the mutated codon for a glutamate residue at position 108 (*Figure 6—figure*

supplement 1 and *Figure 8—figure supplement 1*), whereas the uncut products contained the wild-type codon for valine. Moreover, sequencing of the entire coding region of the Kir6.2 gene in $^{Endo}$mKir6.2$^{V108E}$ mice confirmed the two intended synonymous substitutions in the DNA triplets that encode amino acids threonine 106 and asparagine 107 without any other nucleotide changes. These intended substitutions were designed in the ssODN donor template to prevent re-editing by Cas9 while preserving their original amino acid identities. However, sequencing of the entire coding region of the mutant Kir6.2 transgene in $^{Rosa26}$mKir6.2$^{NDM-V108E}$ mice revealed a single unintended, nonsynonymous substitution in the DNA codon that would lead to an additional N107Y mutation in the translated Kir6.2 protein (*Figure 8—figure supplement 1*). Fortunately, this N107Y mutation had little effect on the affinity of channels for SpTx1 (*Figure 8—figure supplement 2*).

Specific primer pairs were designed using published genomic sequences for the mouse *Kcnj11* gene (Genbank accession: NC_000073.6) or the *Rosa26* locus (Genbank accession: NC_000072.6). Using Primer-BLAST (NIH) on default settings, each candidate pair was checked against the Refseq representative genomes of *Mus musculus* (taxid: 10090) to ensure a single expected PCR product. The oligodeoxynucleotide sequences of the synthesized primers (Sigma-Genosys) and the predicted sizes of the PCR products are given below:

1. For amplifying the entire coding region of the Kir6.2 gene at the endogenous locus, a 1954-bp PCR product was expected with primers 5'-GGTAGACTTATCCCGCCGTG-3' and 5'-TGGGGGCTCAGTAAGCAATG-3'.
2. For amplifying the entire coding region of the Kir6.2 transgene at the *Rosa26* locus, a 1981-bp PCR product was expected with primers 5'-GAGGCTACTGCTGACTCTCAA-3' and 5'-GCTCGTCAAGAAGACAGGGC-3'.
3. For testing the BanII susceptibility, a 760-bp product was expected using PCR product from one or two as the template with primers 5'-CGCCCACAAGAACATTCGAG-3' and 5'-GGTGATGCCCGTGGTTTCTA-3'. Upon exposure to BanII restriction enzyme, the 760-bp PCR product containing the designed V108E mutation would be cut into 565 and 195 bp fragments whereas the wild-type PCR product would remain uncut.
4. For verifying the presence of the Cre gene, a 429-bp PCR product was expected with primers 5'- GCAAGAACCTGATGGACATGTTCAG-3' and 5'-GCAATCCCCAGAAATGCCAGATTAC-3'.

## Preparation of isolated pancreatic islets from mice

Pancreatic islets were isolated as previously described (*Doliba et al., 2017*; *Li et al., 2009*). For each experiment, one to four adult mice (8–12 weeks of age) were euthanized by exposing them to an overdose of isoflurane through inhalation and a subsequent cervical dislocation. Following a perfusion through the common bile duct with a Hank's balanced salt solution (HBSS; GIBCO, 14175) containing collagenase XI (1–2 mg/ml, Millipore-Sigma, C7657) and 1 mM CaCl$_2$, the pancreas was dissected and digested at 37°C for 5 or 15 min. After the digestion, the collagenase in the pancreas homogenate was removed through several washes with HBSS containing 0.3% (wt/vol) bovine serum albumin, and dissociated islets were then hand-picked into a recovery media RPMI1640 (GIBCO, 21870) with the following supplements: 10% fetal bovine serum, 2 mM GlutaMAX, 1 mM sodium pyruvate, penicillin–streptomycin, and 10 mM HEPES titrated to pH 7.4 with NaOH. For the perifusion assay, the digested pancreases samples were subject to a Ficoll gradient purification, and the islets in the enriched fraction were washed with the recovery solution before handpicking islets. During this process, the samples from four mice were pooled to ensure a sufficient number of islets were used to perform assays under compared conditions with the same population in each independent round of experiment. The handpicked islets were placed in 5% CO$_2$ incubator at 37°C to rest overnight prior to the static incubation assay or 2 days prior to the perifusion assay.

## Assays of secreted insulin and total insulin content of mouse islets

For insulin secretion in each independent static incubation assays, pancreatic islets from one mouse (8–12 weeks of age) were used as follows. Isolated islets were washed by six sequential transfers to a modified Krebs–Ringer buffer containing 114 mM NaCl, 24 mM NaHCO$_3$, 5 mM KCl, 2.2 mM CaCl$_2$, 1 mM MgCl$_2$, 1 mM NaH$_2$PO$_4$, 10 mM HEPES titrated to pH 7.4 with NaOH, 0.3% (wt/vol) BSA, and 3 mM glucose. Washed islets were equilibrated in the final wash step for 1 hr at 5% CO$_2$ and 37°C. Using a previous study as reference (*Remedi et al., 2009*) and on the basis of the amount of insulin

secretion and the sensitivity of insulin assay, we chose to use 5–10 islets in each assay. After 1 hr of equilibration, each group of 5–10 randomly selected islets was placed into a well of a microwell plate containing the modified Krebs–Ringer buffer, added with additional test reagents: glucose and SpTx1 as indicated in *Figures 1C and 7E*. Diazoxide (Millipore-Sigma, D9035) was added to the modified Krebs–Ringer buffer 20 min prior to the addition of other test reagents as indicated in *Figure 7E*. Following 1-hr incubation under the conditions of 5% $CO_2$ and 37°C, the supernatant containing released insulin was recovered and the islets were placed in acidified ethanol to extract total insulin content as previously described (*Leiter, 2009*). Insulin in the samples was assayed using a colorimetric ELISA kit (ALPCO, 80-INSRTU-E10). The variations in data values reflected both biological variability and technical errors. No experiments were excluded.

For each perifusion experiment (*Doliba et al., 2017*), pancreatic islets from four mice (8–12 weeks of age) were used in the following manner. About 180 handpicked islets were placed onto a poly-carbonate membrane (Nuclepore, Whatman) inside a polypropylene perifusion chamber (Swinnex, Millipore) and the oxygenated perifusion solution (modified Krebs–Ringer buffer as above) added with additional test reagents: glucose, glibenclamide (Millipore-Sigma, G0639), and/or SpTx1 as indicated in *Figure 7A–D*. All experiments were performed at 37°C with an HPLC-controlled flow rate of 1 ml/min; consecutive fractions of 1 ml were collected. At the end of each experiment, the islets were recovered and placed in acidified ethanol to extract total insulin content as described above. Insulin in the samples was assayed using an insulin-specific radioimmunoassay kit (RI-13K, Millipore). For each condition, the study was performed three to six times. The variations in data values reflected both biological variability and technical errors. No experiments were excluded.

## Administration of tamoxifen and induction of diabetes in NDM-model mice

In $^{Rosa26}$mKir6.2$^{NDM}$ and $^{Rosa26}$mKir6.2$^{NDM-V108E}$ mice, the expression of mutant Kir6.2 activity occurred in the pancreas upon the excision of the Neo/WSS cassette by the tamoxifen-dependent Cre recombinase (*Remedi et al., 2009*). Exposure to tamoxifen permitted the translocation of Cre recombinase from the cytoplasm to the nucleus in order to perform the excision between *loxP* sites within the targeted *Rosa26* locus. To induce the expression of mutant Kir6.2 activity and consequently the diabetic pheno-type in these mutant mice (8–12 weeks of age), corn oil containing tamoxifen (75 mg/kg body weight; prepared as described below) was injected intraperitoneally for five consecutive days. Injection sites were sealed with a tissue adhesive (3M Vetbond) to prevent leakage of the injected liquid.

To make a liquid stock of tamoxifen (20 mg/ml) for injection, tamoxifen solids (Millipore-Sigma, T5648) were added into corn oil (Millipore-Sigma, C8267) in a polypropylene tube wrapped with aluminum foil to protect tamoxifen from light. The stock was placed on a nutator to dissolve overnight at room temperature. The dissolved tamoxifen stock was filtered through a sterilized Steriflip unit (Millipore) then aseptically aliquoted, stored at 4°C, and used within 5 days.

## Mouse blood glucose monitoring and plasma insulin assay

We evaluated the blood glucose levels of tested mice (8–12 weeks of age) using a digital glucometer (Clarity Diagnostics, BG1000). Induced diabetic mice were tested daily starting from the day after the final administration of tamoxifen until their blood glucose levels reached above 600 mg/dl. Following overnight (16 hr) fasting, blood glucose monitoring and plasma collection were then performed using specified mouse lines at designated time points over a 2-hr observation period in the next morning. Glibenclamide (40 mg/kg body weight) or its vehicle pure DMSO was administered through a single intraperitoneal injection whereas SpTx1 (1 mg/kg body weight) or its vehicle normal saline solution was intravenously administered by a lateral tail vein or retro-orbital sinus injection. At each given time point, the glucose level of a blood drop obtained from the cut tail tip was measured twice using a glucometer. For any case where the level was above 600 mg/dl, the blood sample was diluted by an equal volume of the phosphate-buffer solution titrated to pH 7.3 before the glucose level was remeasured for two times. Additionally, four to five drops (~12 µl) of tail vein blood were collected at individual specified time points into a tube containing $K_2$EDTA (1.5 mg/ml blood) and centrifuged at $2000 \times g$ for 5 min at 4°C. 5 µl of the resulting plasma was frozen in liquid nitrogen, stored at −20°C, and assayed using a colorimetric insulin-detecting ELISA kit (CrystalChem, 90080) within a week. Mice were grouped according to genotypes. Using a previous study as reference (*Remedi et al., 2009*),

we determined the number of mice for individual groups. The variations in data values reflected both biological variability and technical errors. No experiments were excluded.

## Statistics

Unless otherwise specified, all data are reported as mean (± SEM). Statistical analyses were performed using software Origin 8.0 (OriginLab) and Igor Pro 8 and 9 (Wavemetrics: https://www.wavemetrics.com). p values were calculated using two-tailed Student's *t*-test with unequal variance (Origin 8.0) and presented in the relevant figure legends purely as descriptive parameters of data.

## Software used to generate figures

The data graphs and statistical analyses in *Figures 1, 3–5*, *Figures 7 and 8* and *Figure 8—figure supplement 2* as well as data values in *Figure 3—figure supplement 1* were created using Origin 8.0 (OriginLab: https://www.originlab.com). Islet cell electrophysiological results in *Figures 2 and 6* were analyzed using Igor Pro 8 and 9. Snippets of the DNA sequencing chromatograms in *Figure 6—figure supplement 1A* and *Figure 8—figure supplement 1A*, B were generated by screen capture, the textual contents of the figure legends were drafted with Word (Microsoft Office Suite version 365; https://www.microsoft.com/en-us/store/apps/windows). All figure panels and legends were made by importing and resizing these vector graphics, image files and texts using Adobe Illustrator (version CS4; https://www.adobe.com).

## Acknowledgements

We thank J Bryan (Baylor College of Medicine) and L Jan (University of California, San Francisco, CA) for sharing Kir6.2 and SUR1 cDNAs; C Nichols (Washington University) for sharing the diabetic model mice; and the University of Pennsylvania Transgenic and Chimeric Mouse Facility for the CRISPR mouse line creation service and the *Diabetes Research Center Core for performing the perifusion and RIA assays of* mouse pancreatic islet β cells (P30-DK19525). The present study in the early phase was supported by the National Institute of Diabetes and Digestive and Kidney Diseases (DK109979) to ZL, and in the later phase when research mice were not directly involved by the National Institute of General Medical Sciences (GM055560) to ZL. TH was in part supported by the National Institute of Diabetes and Digestive and Kidney Diseases (DK098517 and P30-DK19525).

## Additional information

### Funding

| Funder | Grant reference number | Author |
| --- | --- | --- |
| National Institute of Diabetes and Digestive and Kidney Diseases | DK109979 | Zhe Lu |
| National Institute of General Medical Sciences | GM055560 | Zhe Lu |
| National Institute of Diabetes and Digestive and Kidney Diseases | DK098517 | Toshinori Hoshi |
| National Institute of Diabetes and Digestive and Kidney Diseases | P30-DK19525 | Toshinori Hoshi |

The funders had no role in study design, data collection, and interpretation, or the decision to submit the work for publication.

### Author contributions

Yajamana Ramu, Jayden Yamakaze, Yufeng Zhou, Data curation, Formal analysis, Investigation, Methodology, Validation, Writing – review and editing; Toshinori Hoshi, Data curation, Formal analysis, Funding acquisition, Investigation, Methodology, Resources, Validation, Writing – review and editing;

Zhe Lu, Conceptualization, Formal analysis, Funding acquisition, Methodology, Resources, Supervision, Writing – original draft, Writing – review and editing

**Author ORCIDs**
Zhe Lu http://orcid.org/0000-0001-7108-9303

**Ethics**
The Institutional Animal Care and Use Committee at the University of Pennsylvania reviewed and approved this study (Protocol Numbers: 804489 and 804811). All animal welfare considerations were taken and the study was performed in strict accordance with the recommendations in the Guide for the Care and Use of Laboratory Animals of the National Institutes of Health. All animal care was provided by experienced staff and researchers, and tissue dissections were performed after euthanasia by researchers who were trained at the University Laboratory Animal Resources of the University of Pennsylvania. Every effort was made to minimize suffering and distressed of mice. The studies were performed in compliance with the ARRIVE guidelines (https://arriveguidelines.org/).

**Decision letter and Author response**
Decision letter https://doi.org/10.7554/eLife.77026.sa1
Author response https://doi.org/10.7554/eLife.77026.sa2

## Additional files

**Supplementary files**
• Transparent reporting form

**Data availability**
Datasets generated or analyzed in the present study are included in the manuscript and supporting files. Source data files are provided for Figures 1-8 and the associated figure supplement files.

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
