## [Editor Report]

ATP-sensitive potassium channels (K_ATP_ channels) play a key role in glucose-stimulated insulin secretion from pancreatic β cells, and they are a major therapeutic target in diabetes mellitus. This paper provides clear evidence in mice that the effects of a peptide toxin that blocks the pore-forming Kir6.2 subunit are subtly but importantly different from the effects of sulfonylureas that act on an auxiliary subunit of the K_ATP_ channels. The pore-blocking peptide toxin has effects on β cells and on mice only when they contain a mutant channel that is sensitive to the toxin, unlike the native mouse channel; in contrast, the sulfonylurea glibenclamide appears to have additional downstream effects on insulin secretion, even in the absence of the normal glucose stimulus.

---

## [Decision Letter]

**Decision letter after peer review:**

[Editors’ note: the authors submitted for reconsideration following the decision after peer review. What follows is the decision letter after the first round of review.]

Thank you for submitting your work entitled "Blocking Kir6.2 channels boosts glucose-induced insulin secretion from β cells and lowers blood glucose in diabetic mice" for consideration by *eLife*. Your article has been reviewed by 3 peer reviewers, and the evaluation has been overseen by a Reviewing Editor and a Senior Editor. The following individual involved in review of your submission has agreed to reveal their identity: Kenton J Swartz (Reviewer #2).

Our decision has been reached after consultation between the reviewers. Based on these discussions and the individual reviews below, we regret to inform you that your work will not be considered further for publication in *eLife*.

The paper presents interesting work on a high-affinity toxin for the pore-forming Kir6.2 subunit of the K_ATP_ channel. Because the K_ATP_ channel is a target for therapy of diabetes mellitus by sulfonylurea drugs that inhibit the channel less directly, this toxin is an interesting and potentially valuable tool for improving diabetes therapy. The authors have produced a mouse line that exhibits the high-affinity blockade seen for the human channel, and have compared the action of SpTx1 and the sulfonylurea glibenclamide on insulin secretion from pancreatic islets in vitro, and on blood glucose in vivo in a mouse model of permanent neonatal diabetes mellitus (PNDM).

The results are effectively in three parts, and (as summarized here and explained in the detailed reviews below), the reviewers had concerns about the latter two.

1) Identification of mutations needed to make the mouse channel susceptible to high affinity SpTx1 blockade, and construction of the mouse with the knock-in mutation. The reviewers all agreed that this is solid and valuable.

2) Studies of glibenclamide and SpTx1 effects on insulin secretion in islets from the mouse. These results are interesting at a superficial level – the difference between the drugs in what the authors term "stimulation" vs. "potentiation" – but the basis of the difference is unexplored and somewhat mysterious. It is neither explained by the known effect of glib on secretion from depolarized islets, nor by the known state-dependence of sulfonylurea effects. It is unclear whether there is a difference in the drugs' effect on β cell membrane potential or rather on a later step of secretion. Off-target effects of SpTx1 could be an alternative explanation, and comparison of the wt and mutant mouse could clarify this.

3) The in vivo test of the SpTx1-sensitive-plus-PNDM mouse. There are a number of problems with these experiments, as detailed in the reviews, and alternative approaches would be much better (as well as doing the glibenclamide vs. SpTx1 experiments in the same line of mice).

*Reviewer #1:*

This paper describes some biological effects of the authors' previously reported toxin against the Kir6.2 channel, which provides the pore component of K_ATP_ channels in pancreatic β cells as well as other cells.

The SpTx1 toxin has much higher affinity for the human Kir6.2 pore than for the mouse Kir6.2, and the authors identify a specific point mutation (of the three sequence differences near the pore) that can interconvert the affinities. They then make this sensitivity-conferring mutation in two mouse lines, one with nominally normal K_ATP_ channels and another with an inducible transgenic with a chronically-active mutant Kir6.2 channel.

With these new mouse tools, they show that in isolated islets from "normal" mice with toxin sensitivity, doses of SpTx1 and the SUR1-acting drug glibenclamide that produce approximately the same effect on peak glucose-stimulated insulin secretion have slightly different effects on baseline (pre-glucose-challenge) secretion. They attribute this quantitative difference to an off-target effect of glibenclamide, but the evidence is weak as only one dose of glibenclamide and two of SpTx1 were examined.

They also show a slightly more potent reversal by SpTx1 than glibenclamide of the hyperglycemia in the mice with chronically-active Kir6.2 channels. This probably indicates that the affinity and pharmacodynamics of the injected toxin allow for better channel inhibition than with glibenclamide.

Although this possible improvement in pharmacology might be useful in extreme cases (despite the barriers to protein-based therapies as clinical drugs), it is hard to see that this is a clear general improvement, or that new biological understanding has emerged from this study.

*Reviewer #2:*

This is an interesting ms from the Lu laboratory that explores whether a pore-blocking centipede toxin (SpTx) targeting the mouse K_ATP_ channel in pancreatic β cells can modulate insulin secretion from mouse pancreatic islets and in live mice. The role of K_ATP_ in controlling insulin secretion is well established and a widely used class of antidiabetic sulfonylurea drugs inhibit KATP by binding to the regulatory sulfonylurea receptor (SUR). Although sulfonylureas are widely used to treat diabetes, they can have side effects, can lose efficacy with chronic usage and many mutations causing permanent neonatal diabetes mellitus (PNDM) render sulfonylureas ineffective. Moreover, sulfonylureas may target other proteins involved in insulin secretion and the role of K_ATP_ in the plasma membrane of β cells has not been tested by directly blocking the pore of the channel. The authors begin by characterizing the affinity of SpTx1 for mouse K_ATP_ formed by expressing mKir6.2 and mSUR1 and find that the toxin inhibits this channel with much lower affinity (~700 nM) compared to the human channel (~15 nM) formed by the equivalent subunits, representing a problem for undertaking studies on insulin secretion using mouse models. They undertake a systematic evaluation and show using multiple approaches that the lower affinity for mK_ATP_ is not due to mSUR1, but the results of one residue difference in the outer vestibule of Kir6.2, where the human channel has a Glu and the mouse channel has a Val. They then engineer a mouse with the V108E mutation in Kir6.2 using CRISPR-Cas9 and examine whether SpTx1 promotes insulin secretion from isolated pancreatic islets and diminishes blood glucose live animals. The authors find that SpTx effectively promotes insulin secretion from islets when stimulated with 10 mM glucose, but tends not to promote insulin secretion at basal levels of glucose (3 mM) unless applied at much higher concentrations. These effects contrast with the sulfonylurea glibenclamide, which enhances both basal and stimulated insulin secretion. Using a mouse model of PNDM that they cross with the Kir6.2 V108E mouse, they show that SpTx1 lowers blood glucose more effectively than glibenclamide. Overall, this is a clearly presented, well-controlled and rigorous study demonstrating that a pore-blocking toxin can inhibit K_ATP_ to promote insulin secretion and lower blood glucose, providing confirmation of the critical role of K_ATP_ in regulating insulin secretion and demonstrating that pore-blockers of this channel could be useful for treating diabetes. The distinct features of pore-blockade compared to sulfonylureas, in particular for basal insulin secretion, will provide alternatives to sulfonylureas with fewer side effects and pore-blockers would be viable options for PNDM cases where sulfonylureas are not effective. The study also elegantly demonstrates that sulfonylureas influence insulin secretion though mechanisms not solely involving K_ATP_ in the plasma membrane of pancreatic β cells. This study will make a cool *eLife* paper and I have only a few suggestions for revision.

1) The lack of effect of SpTx1 in promoting insulin secretion from islets obtained from wt mice (even up to 1 mM!!) is states as being not surprising, which is true given the lower sensitivity of the mouse K_ATP_ channel to SpTx1. However, this experiment is also a critical control demonstrating that SpTx1 acts specifically on K_ATP_ in the V108E mouse to promote insulin secretion and lower blood glucose. I suggest the authors move this result to the main ms after Figure 6 so its full impact is clear to the readers. It would also be good to convert it to a scatter plot with mean and error bars without connecting lines as there is no effect at the concentrations tested. The authors might also want to consider using color in more of their figures as *eLife* does not charge for color figures.

2) I am curious whether SpTx1 was well-tolerated by mice. I understand these are proof-of-concept experiments and the authors have not done formal behavioral or toxicity studies, I'm just curious if the authors notice anything interesting.

3) In Figures1-4, important control data from earlier figures and panels are replotted multiple times for comparison, but this could give the unintended impression that experiments were redone for each of the subsequent panels. It would be good to use a colored function obtained by fitting the equation to the data the first time and then to simple replot that function without data points so the reader can compare. It would also be helpful to add a table with SpTx1 Kd values to facilitate comparison of the large number of measurements reported here.

*Reviewer #3:*

This interesting paper describes the effect of a novel Kir6.2-specific pore blocker, SpTx1, on insulin secretion. It clearly shows that human Kir2 is more sensitive to SpTx1 than mouse Kir6.2 due to a single residue (E108 in humans, V108 in mice). It reports a novel Crispr mouse model containing the V108K mutation and shows that SpTx1 stimulates secretion from isolated islets and lowers blood glucose in an OGTT. The patch clamp experiments are clear and well conducted. However, the secretion experiments could be better described, the OGTT is not very informative, and it is unclear why SpTx1 does not stimulate insulin secretion at subthreshold (3mM) glucose. The paper could be written with greater clarity and the authors overplay some aspects of the study and downplay others.

A numbered summary of any substantive concerns:

1) The authors fail to mention that sulphonylureas like glibenclamide block the channel in a very complex fashion – their efficacy is influenced by the metabolic regulation of the channel by adenine nucleotides. In the absence of nucleotides, sulphonylureas never block the channel completely, but only by about 60-70%. In the presence of nucleotides, sulphonylureas displace MgADP binding, preventing MgADP activation and unmasking ATP inhibition at Kir6.2. This causes an apparent enhancement of sulphonylurea block, such that wild-type channels can be almost fully blocked by a maximal concentration of sulphonylurea. This interaction with nucleotides means sulphonylurea block is sensitive to the metabolic state of the cell. It also that drugs carrying PNDM mutations that strongly reduce the inhibitory effect of ATP will never be fully blocked by sulphonylureas even at maximal concentrations (Proks et al. Diabetes 62, 2909). An important point about SpTx1 action is that (I assume) it is independent of metabolic state of the cell or of the ATP sensitivity of Kir6.2. The authors should mention this somewhere in the text.

2) The authors should state that another very important reason for searching for a specific Kir6.2 blocker is that the K_ATP_ channel is expressed in brain neurones and as a consequence many PNDM patients with Kir6.2 mutations have neurological problems, some of them very severe. Sulphonylureas have very limited effects on these problems, probably because they are rapidly pumped out of the brain. A drug that blocks overactive neuronal channels would be very valuable. Do the authors know whether their drug gets into the brain? If it does, some of the in vivo effects on glucose homeostasis could be mediated via a central effect. This should be discussed.

3) The reduced sulphonylurea sensitivity of PNDM channels is overplayed and should be toned down. While it is true that some patients with rare and very severe mutations cannot be transferred to sulphonylurea therapy (and need a different therapy) this is a very small number. Most PNDM patients do very well on sulphonylureas.

4) Reduced effectiveness of sulphonylureas. This is true but the phrasing might be misleading. There are many reasons why PNDM patients need a higher dose than type 2 diabetes patients, including, 1) their K_ATP_ channels are more open. 2) the ability of sulphonylureas to block the channel is dependent on the ATP sensitivity of the channel (see below). I am not aware that there are any PNDM mutations (to date) that influence the binding of sulphonylureas.

5) While it is true that a major side effect of sulphonylurea therapy in type 2 diabetes is hypoglycaemia, it turns out that drug-induced hypoglycaemia in PNDM patients is rare (Bowman et al., Lancet Diabetes Endocrinol. 6, 637-646). So I think it is wrong to imply that SpTx1 is preferable to sulphonylureas in PNDM because it is likely to cause less hypoglycaemia.

6) There is no clear explanation for why SpTx1 does not work at low glucose. Blocking the K_ATP_ channel should lead to membrane depolarisation and calcium entry and so stimulate secretion. Does this in fact occur at 3mM glucose? I think it important to perform perforated patch studies on β-cells from Rosa26mKir.2V108E mice and look at electrical activity at high SpTx1 concentrations (e.g. 2µM) in 3 and 10mM glucose. Does it in fact depolarise the cell as the authors suppose? Another, and perhaps easier, approach would be to do calcium imaging. Is calcium increased by SpTx1 in intact β cells at low glucose? Could SpTx1 block not only Kir6.2 but also the background inward current that mediates depolarisation when K_ATP_ is blocked?

7) A problem with glibenclamide is that it is lipid soluble, and it tends to accumulate inside the β-cell. This may be one reason for its slow off-rate in the secretion studies. The other is that it unbinds slowly from the receptor so that the K_ATP_ current never recovers fully on the time scale of an electrophysiological experiment (ref). The authors suggest this is a problem and a fast reversible blocker is needed. However, there are other sulphonylureas (like gliclazide), which are high affinity blockers with a fast off rate.

8) Overall, I don’t think there is a need to state sulphonylureas are inadequate in order to justify the development of SpTx1. The toxin is an important step forward in itself.

9) If it is really the case that SpTx1 has no effect at sub-stimulatory glucose concentrations, the authors might have found the pharmaceutical dream goal – a drug that is effective in type 2 diabetes only in response to a meal and not at resting glucose levels. A lot more work is needed to show that, and I don’t think the authors need to do that for this paper – but they might want to mention it more clearly in the Discussion as a possibility that they will explore in the future.

10) Glucose tolerance tests are not very helpful in severely diabetic mice as the blood glucose levels are already so high and usually off the glucometer scale. More valuable would be to see what happens to the fasting (or free fed) blood glucose in drug treated mice. This could be done by implantation of a subcutaneous slow-release pellet (or an osmotic minipump) and following blood glucose for a couple of days.

---

## [Author Response]

[Editors’ note: the authors resubmitted a revised version of the paper for consideration. What follows is the authors’ response to the first round of review.]

The paper presents interesting work on a high-affinity toxin for the pore-forming Kir6.2 subunit of the K_ATP_ channel. Because the K_ATP_ channel is a target for therapy of diabetes mellitus by sulfonylurea drugs that inhibit the channel less directly, this toxin is an interesting and potentially valuable tool for improving diabetes therapy. The authors have produced a mouse line that exhibits the high-affinity blockade seen for the human channel, and have compared the action of SpTx1 and the sulfonylurea glibenclamide on insulin secretion from pancreatic islets in vitro, and on blood glucose in vivo in a mouse model of permanent neonatal diabetes mellitus (PNDM).The results are effectively in three parts, and (as summarized here and explained in the detailed reviews below), the reviewers had concerns about the latter two.1) Identification of mutations needed to make the mouse channel susceptible to high affinity SpTx1 blockade, and construction of the mouse with the knock-in mutation. The reviewers all agreed that this is solid and valuable.

We keep these parts of the manuscript largely as they were.

2) Studies of glibenclamide and SpTx1 effects on insulin secretion in islets from the mouse. These results are interesting at a superficial level – the difference between the drugs in what the authors term “stimulation” vs. “potentiation” – but the basis of the difference is unexplored and somewhat mysterious. It is neither explained by the known effect of glib on secretion from depolarized islets, nor by the known state-dependence of sulfonylurea effects. It is unclear whether there is a difference in the drugs’ effect on β cell membrane potential or rather on a later step of secretion. Off-target effects of SpTx1 could be an alternative explanation, and comparison of the wt and mutant mouse could clarify this.

Regarding off-target effects, SpTx1 did not affect glucose-stimulated insulin secretion (GSIS) from isolated islets of wild-type mice, but it did act as an effective secondary secretagogue that strongly potentiates GSIS from those of the SpTx1-senstive mice. In addition, we now show that SpTx1 could effectively counteract the inhibition on GSIS by the K_ATP_ opener diazoxide.

Additionally, we now include membrane potential measurements recorded in the perforated whole-cell mode from individual β cells near the surface of isolated but intact islets. SpTx1 did not depolarize the membrane of β cells of wild-type mice, but it did depolarize the membrane of β cells of SpTx1-senstive-mutant mice under certain glucose conditions. Thus, in the present context, SpTx1 does not have any impactful off-target effects. In regard to the differing effects of SpTx1 and glibenclamide on insulin secretion in subthreshold 3 mM and stimulating concentrations of 10 mM glucose, we observed that SpTx1 did not depolarize the β cell membrane in the absence of glucose. In 5 mM glucose, it had variable effects on the β cell membrane potential and elicited no action potentials whereas glibenclamide caused depolarization more reliably, and triggered action potentials. In 8 mM glucose, SpTx1 also caused depolarization more consistently and triggered action potentials. Thus, SpTx1 apparently cannot mimic some effects of glibenclamide well, or cannot at all, which most likely occur downstream of the K_ATP_ channels, as suggested by reviwer#3.

3) The in vivo test of the SpTx1-sensitive-plus-PNDM mouse. There are a number of problems with these experiments, as detailed in the reviews, and alternative approaches would be much better (as well as doing the glibenclamide vs. SpTx1 experiments in the same line of mice).

As suggested by reviewer #3, we now present the in vivo data on the fasted SpTx1-sensitive NDM-model mice without glucose challenge, instead of those previous data obtained under a glucose-challenged condition. In addition, we show the corresponding plasma insulin levels along with the blood glucose levels. Also, the glibenclamide and SpTx1 in vivo experiments were performed with same ^Rosa26^mKir6.2^NDM-V108E^ mouse line. However, the glibenclamide experiment is merely carried out to show that ^Rosa26^mKir6.2^NDM-V108E^ mouse line retains the expected ineffectiveness of glibenclamide in lowering elevated blood glucose in the original ^Rosa26^mKir6.2^NDM^ mouse line from the Nichols’ group, as they reported.

Reviewer #1:This paper describes some biological effects of the authors' previously reported toxin against the Kir6.2 channel, which provides the pore component of K_ATP_ channels in pancreatic β cells as well as other cells.The SpTx1 toxin has much higher affinity for the human Kir6.2 pore than for the mouse Kir6.2, and the authors identify a specific point mutation (of the three sequence differences near the pore) that can interconvert the affinities. They then make this sensitivity-conferring mutation in two mouse lines, one with nominally normal K_ATP_ channels and another with an inducible transgenic with a chronically-active mutant Kir6.2 channel.With these new mouse tools, they show that in isolated islets from "normal" mice with toxin sensitivity, doses of SpTx1 and the SUR1-acting drug glibenclamide that produce approximately the same effect on peak glucose-stimulated insulin secretion have slightly different effects on baseline (pre-glucose-challenge) secretion. They attribute this quantitative difference to an off-target effect of glibenclamide, but the evidence is weak as only one dose of glibenclamide and two of SpTx1 were examined.

We now plot in Figure 7D the ratios between the amounts of insulin secretion in the presence of SpTx1 and glibenclamide along the glucose concentration steps. These plots clearly show that ratio varied between two tested glucose concentrations. The data suggest that in high (16.7 mM) glucose, SpTx1 is as a good secondary insulin secretagogue as glibenclamide is whereas it is not as a good primary insulin secretagogue in subthreshold (3 mM) glucose. Thus, SpTx1 apparently cannot mimic some effects of glibenclamide well, or cannot at all. As suggested by reviewer #3, one possibility is SpTx1 apparently cannot mimic some effects of glibenclamide in low glucose concentrations well, or cannot at all, which occur. e.g., downstream of the K_ATP_ channels. This possibility predicts that SpTx1 would depolarize the cell membrane in high but not low glucose concentrations.

As mentioned under item #2 of editors’ summaries, we now show that the effect of SpTx1 on the membrane potential of β cells of SpTx1-senstive-mutant mice is indeed glucose concentration dependent. SpTx1 causes little depolarization of the β cell membrane potential in the absence of glucose. In 5 mM glucose, it had variable effects on the β cell membrane potential and elicited no action potentials whereas glibenclamide caused depolarization more constantly, and triggered action potentials. In 8 mM glucose, SpTx1 caused depolarization more consistently, and also triggered action potentials.

To understand the exact molecular mechanism underlying the observed difference between the inhibitors undoubtedly requires a major effort beyond the present scope.

They also show a slightly more potent reversal by SpTx1 than glibenclamide of the hyperglycemia in the mice with chronically-active Kir6.2 channels. This probably indicates that the affinity and pharmacodynamics of the injected toxin allow for better channel inhibition than with glibenclamide.

Glibenclamide’s effect on K_ATP_ channel inhibition is known to wash-off slowly, and we previously showed that the off-rate of SpTx1 is much more rapid (0.12 s^-1^). Thus, as the reviewer suggests, a better inhibition by glibenclamide is expected to occur during the wash-out period, but not when both glibenclamide and SpTx1 are maintained in sufficiently high steady-state concentrations. In any case, the difference in the apparent “off-kinetics” of SpTx1 and glibenclamide is remarkable. Give this difference is not our focus, we leave them as observations and no longer comment on them.

Although this possible improvement in pharmacology might be useful in extreme cases (despite the barriers to protein-based therapies as clinical drugs), it is hard to see that this is a clear general improvement, or that new biological understanding has emerged from this study.

We developed SpTx1 foremost as an experimental tool. One of its usages showcased here is to identify potential effects of targeting Kir6.2 itself in a proof-of-concept study. We do not suggest using SpTx1 as a therapeutic drug. We do not know a precedence that a high-affinity and relatively specific toxin inhibitor for a biologically important channel, which is effective both *in intro* and in vivo, is not an important development.

Reviewer #2:[…]1) The lack of effect of SpTx1 in promoting insulin secretion from islets obtained from wt mice (even up to 1 mM!!) is states as being not surprising, which is true given the lower sensitivity of the mouse K_ATP_ channel to SpTx1. However, this experiment is also a critical control demonstrating that SpTx1 acts specifically on K_ATP_ in the V108E mouse to promote insulin secretion and lower blood glucose. I suggest the authors move this result to the main ms after Figure 6 so its full impact is clear to the readers. It would also be good to convert it to a scatter plot with mean and error bars without connecting lines as there is no effect at the concentrations tested. The authors might also want to consider using color in more of their figures as eLife does not charge for color figures.

Following the reviewer’s suggestions, we now present those results as a scatter plot with colors in Figure 1C and incorporate more colors in the other figures. We also more clearly emphasize that the lack of an effect of SpTx1 on insulin secretion from isolated islets of wild-type mice is a control for potential off-target effects. “In this concentration range, SpTx1 failed to alter insulin secretion in the presence of either 7 mM or 15 mM glucose (Figure 1C). Thus, SpTx1 had no significant effects on GSIS either through acting on mKir6.2 or any unintended targets within the wild-type mouse islets.”

2) I am curious whether SpTx1 was well-tolerated by mice. I understand these are proof-of-concept experiments and the authors have not done formal behavioral or toxicity studies, I'm just curious if the authors notice anything interesting.

We now mention in the Results: “SpTx1 at a dose of 1 mg/kg neither had discernible effect on the blood glucose levels of wild-type mice (Figure 1B) nor caused other noticeable differences during the observation period, when compared to wild-type mice administered with the vehicle saline.”

3) In Figures1-4, important control data from earlier figures and panels are replotted multiple times for comparison, but this could give the unintended impression that experiments were redone for each of the subsequent panels. It would be good to use a colored function obtained by fitting the equation to the data the first time and then to simple replot that function without data points so the reader can compare. It would also be helpful to add a table with SpTx1 Kd values to facilitate comparison of the large number of measurements reported here.

We have implemented the reviewer’s suggestions in the revised version.

Reviewer #3:A numbered summary of any substantive concerns:1) The authors fail to mention that sulphonylureas like glibenclamide block the channel in a very complex fashion – their efficacy is influenced by the metabolic regulation of the channel by adenine nucleotides. In the absence of nucleotides, sulphonylureas never block the channel completely, but only by about 60-70%. In the presence of nucleotides, sulphonylureas displace MgADP binding, preventing MgADP activation and unmasking ATP inhibition at Kir6.2. This causes an apparent enhancement of sulphonylurea block, such that wild-type channels can be almost fully blocked by a maximal concentration of sulphonylurea. This interaction with nucleotides means sulphonylurea block is sensitive to the metabolic state of the cell. It also that drugs carrying PNDM mutations that strongly reduce the inhibitory effect of ATP will never be fully blocked by sulphonylureas even at maximal concentrations (Proks et al. Diabetes 62, 2909). An important point about SpTx1 action is that (I assume) it is independent of metabolic state of the cell or of the ATP sensitivity of Kir6.2. The authors should mention this somewhere in the text.

We now state in the Discussion: “As experimental tools, sulfonylureas and SpTx1, while both inhibiting K_ATP_ channels, have important and different characteristics. The pancreatic K_ATP_ channels made of Kir6.2 and SUR1 are inhibited by ATP and activated by MgADP (Gribble and Reimann, 2003; Proks et al., 2013). The stimulatory effect of MgADP is antagonized by binding of sulfonylureas to SUR1. This apparent inhibitory effect of sulfonylureas, together with the ATP-mediated inhibition, decreases the channel open probability (P_o_) to near zero. However, without ATP, sulfonylureas alone can only inhibit about 2/3 of the channel current. Thus, the inhibitory effect of sulfonylureas depends on the cellular metabolic state. In contrast, the inhibitory effect of SpTx1 on the Kir6.2 pore in the K_ATP_ channel complex is not expected to depend on the cellular metabolic state. This expectation stems from the observation that SpTx1 blocks, comparably well, currents through a constitutively active Kir6.2 mutant and those through K_ATP_ channels activated with azide that lowers the concentration of intracellular ATP (Figures 3-5).”

2) The authors should state that another very important reason for searching for a specific Kir6.2 blocker is that the K_ATP_ channel is expressed in brain neurones and as a consequence many PNDM patients with Kir6.2 mutations have neurological problems, some of them very severe. Sulphonylureas have very limited effects on these problems, probably because they are rapidly pumped out of the brain. A drug that blocks overactive neuronal channels would be very valuable. Do the authors know whether their drug gets into the brain? If it does, some of the in vivo effects on glucose homeostasis could be mediated via a central effect. This should be discussed.

We now mention in the Discussion: “Because SpTx1 and sulfonylureas have different characteristics, these inhibitors may have different therapeutic potentials. For example, sulfonylureas have only limited effects on the neurological disorders of patients with DEND (Hattersley and Ashcroft, 2005; Pipatpolkai et al., 2020). To this end, it will be interesting to test whether SpTx1 can help to mitigate these neurological disorders in animal models by directly delivering it into the cerebrospinal fluid.”

We do not expect that SpTx1 can effectively get across the blood-brain barrier.

3) The reduced sulphonylurea sensitivity of PNDM channels is overplayed and should be toned down. While it is true that some patients with rare and very severe mutations cannot be transferred to sulphonylurea therapy (and need a different therapy) this is a very small number. Most PNDM patients do very well on sulphonylureas.

We followed the reviewer’s advice, and we no longer emphasize this issue.

4) Reduced effectiveness of sulphonylureas. This is true but the phrasing might be misleading. There are many reasons why PNDM patients need a higher dose than type 2 diabetes patients, including, 1) their K_ATP_ channels are more open. 2) the ability of sulphonylureas to block the channel is dependent on the ATP sensitivity of the channel (see below). I am not aware that there are any PNDM mutations (to date) that influence the binding of sulphonylureas.

We now merely present the glibenclamide experiment to show that ^Rosa26^mKir6.2^NDM-V108E^ mouse line retains the expected relative ineffectiveness of glibenclamide in lowering elevated blood glucose in the original ^Rosa26^mKir6.2^NDM^ mouse line from the Nichols’ group, as they reported.

5) While it is true that a major side effect of sulphonylurea therapy in type 2 diabetes is hypoglycaemia, it turns out that drug-induced hypoglycaemia in PNDM patients is rare (Bowman et al., Lancet Diabetes Endocrinol. 6, 637-646). So I think it is wrong to imply that SpTx1 is preferable to sulphonylureas in PNDM because it is likely to cause less hypoglycaemia.

We no longer emphasize the point that sulfonylureas can cause hypoglycemia in Discussion. Instead, on one hand, we show that at the indicated dose, glibenclamide did expectedly lower blood glucose levels of fasted wild-type mice by about half during the observed period but did not markedly lower highly elevated glucose levels in the original ^Rosa26^mKir6.2^NDM^. On the other hand, SpTx1 did not lower blood glucose levels of fasted wild-type or ^Endo^Kir6.2^V108E^ mice but did markedly lower highly elevated glucose levels in ^Rosa26^mKir6.2^NDM-V108E^ at the indicated dose during the observed period.

6) There is no clear explanation for why SpTx1 does not work at low glucose. Blocking the K_ATP_ channel should lead to membrane depolarisation and calcium entry and so stimulate secretion. Does this in fact occur at 3mM glucose? I think it important to perform perforated patch studies on β-cells from Rosa26mKir.2V108E mice and look at electrical activity at high SpTx1 concentrations (e.g. 2µM) in 3 and 10mM glucose. Does it in fact depolarise the cell as the authors suppose? Another, and perhaps easier, approach would be to do calcium imaging. Is calcium increased by SpTx1 in intact β cells at low glucose? Could SpTx1 block not only Kir6.2 but also the background inward current that mediates depolarisation when K_ATP_ is blocked?

We now show that at SpTx1 at 0.2 µM (10-fold of its K_d_) depolarized the membrane of β cells within isolated islets from ^Endo^Kir6.2^V108E^ mice consistently in 8 mM glucose, triggering action potentials, and variably in 5 mM glucose without triggering any action potentials though, but little in 0 mM glucose. SpTx1 at 0.2 µM did not depolarize β cells in islets from wild-type mice under these all three glucose concentrations. Regarding the possibility that SpTx1 blocks the background inward current, we did not observe statistically significant hyperpolarization caused by SpTx1 in wild-type β cells (Figure 2).

7) A problem with glibenclamide is that it is lipid soluble, and it tends to accumulate inside the β-cell. This may be one reason for its slow off-rate in the secretion studies. The other is that it unbinds slowly from the receptor so that the K_ATP_ current never recovers fully on the time scale of an electrophysiological experiment (ref). The authors suggest this is a problem and a fast reversible blocker is needed. However, there are other sulphonylureas (like gliclazide), which are high affinity blockers with a fast off rate.

We no longer comment on the biological consequence of apparent kinetics of glibenclaimide.

8) Overall, I don't think there is a need to state sulphonylureas are inadequate in order to justify the development of SpTx1. The toxin is an important step forward in itself.

We heeded the reviewer’s advice and made appropriate adjustments, as quoted under comment#1.

9) If it is really the case that SpTx1 has no effect at sub-stimulatory glucose concentrations, the authors might have found the pharmaceutical dream goal – a drug that is effective in type 2 diabetes only in response to a meal and not at resting glucose levels. A lot more work is needed to show that, and I don't think the authors need to do that for this paper – but they might want to mention it more clearly in the Discussion as a possibility that they will explore in the future.

We now show the effect of SpTx1 on the fasting blood glucose and the plasma insulin levels of diabetic and non-diabetic mice.

10) Glucose tolerance tests are not very helpful in severely diabetic mice as the blood glucose levels are already so high and usually off the glucometer scale. More valuable would be to see what happens to the fasting (or free fed) blood glucose in drug treated mice. This could be done by implantation of a subcutaneous slow-release pellet (or an osmotic minipump) and following blood glucose for a couple of days.

We now show the effect of SpTx1 on the fasting blood glucose and the plasma insulin levels of diabetic and non-diabetic mice.